# Multi-Objective Bayesian Optimization via Adaptive $\varepsilon$-Constraints Decomposition

**Yaohong Yang** [1 2]  **Sammie Katt** [1 2]  **Samuel Kaski** [1 2 3]

## Abstract

Multi-objective Bayesian optimization (MOBO) provides a principled framework for optimizing multiple expensive black-box functions. However, existing MOBO methods often struggle with coverage, scalability, and handling constraints and preferences. In this work we propose *STAGE-BO, Sequential Targeting Adaptive Gap-Filling $\varepsilon$-Constraint Bayesian Optimization*: by analyzing the coverage of the surrogate Pareto front, our method identifies the Pareto front point with the largest uncovered gap, and uses its coordinates to define adaptive constraints in $\varepsilon$-constraint method, which transforms the problem into a sequence of inequality-constrained subproblems, efficiently solved via constrained expected improvement acquisition. Our approach provides uniform Pareto coverage without hypervolume computation and naturally handles constraints and preferences. Experiments on synthetic and real-world benchmarks demonstrate superior coverage and competitive hypervolume performance against state-of-the-art baselines. Our code implementation can be found at https://github.com/YangYaohong1/STAGE-BO.

## 1. Introduction

Multi-objective Bayesian optimization (MOBO) has emerged as a powerful paradigm for optimizing multiple expensive black-box functions (Daulton et al., 2020; Tu et al., 2022; Ngo et al., 2025). By navigating trade-offs, MOBO facilitates discovery in diverse fields ranging from machine learning (Sener & Koltun, 2018), materials science (Xu et al., 2025) to robotics (Kouritem et al., 2022).

[1]Department of Computer Science, Aalto University, Espoo, Finland [2]ELLIS Institute Finland [3]Department of Computer Science, University of Manchester, Manchester, United Kingdom. Correspondence to: Yaohong Yang <yaohong.yang@aalto.fi>.

*Proceedings of the 43rd International Conference on Machine Learning*, Seoul, South Korea. PMLR 306, 2026. Copyright 2026 by the author(s).

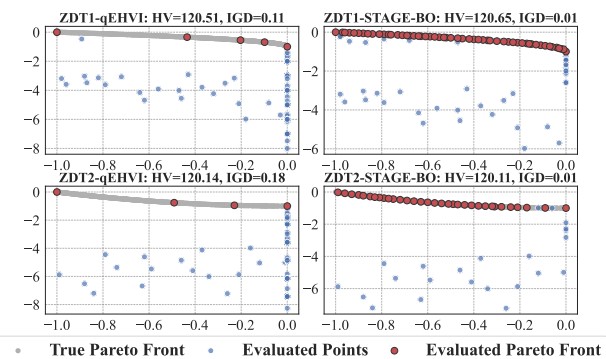

*Figure 1.* **Pareto front approximation on ZDT1 and ZDT2 benchmarks.** While qEHVI (left) and our method (right) achieve comparable hypervolume (HV), ours yields an order-of-magnitude reduction in inverse generational distance (IGD), demonstrating significantly better uniform coverage of the Pareto front. This motivates the need for multiple metrics to assess solution quality.

The objective functions in MOBO are often conflicting, meaning that improving one objective may deteriorate another. Hence the goal is not to identify a single optimal solution, but rather a set of *Pareto optimal solutions*. The standard metric for evaluating Pareto approximations is hypervolume (HV), which measures the size of the dominated objective space. Because HV is strictly Pareto-compliant, the majority of MOBO algorithms aim to maximize HV improvement. However, this reliance on HV introduces two limitations. First, **computational scalability:** exact HV computation scales exponentially with the number of objectives, making it prohibitively expensive for the problem with more than four objectives. Second, **geometric bias:** analysis by Auger et al. (2009) shows that asymptotically, the density of solutions maximizing HV is proportional to the square root of the Pareto front's negative slope ($\propto \sqrt{-F'(\mathbf{x})}$, where $F = [f_1, \ldots, f_m]$) are the objectives). Consequently, HV maximization exhibits a bias: it heavily concentrates solutions in steep *knee* regions while under-sampling *flat* trade-off areas, where the slope is small or close to zero, failing to uniformly cover the front.

Empirical evidence supports this theoretical concern. As illustrated in Figure 1, we observe that while qEHVI (Daulton et al., 2020) achieves high HV scores, it fails to generate a uniformly diverse set of solutions. Thus its inverse gener-

ational distance (IGD)—the distance from the true Pareto front to the obtained Pareto front—can be an order of magnitude larger than that of our proposed method. This discrepancy underscores the limitation of relying solely on HV for evaluation and highlights the necessity of using multiple metrics, such as HV and IGD, to comprehensively assess Pareto front quality in the context of algorithm design.

Diversity-guided MOBO methods attempt to improve coverage but often face significant bottlenecks: they either rely on input-space metrics (Renganathan & Carlson, 2025) that do not guarantee diversity in the objective space, or retain the computational cost of HV maximization (Konakovic Lukovic et al., 2020; Ngo et al., 2025; Ahmadianshalchi et al., 2024). Scalarization methods (Paria et al., 2020; Knowles, 2006) avoid these costs by decomposing the problem via random weights. However, it is well-established that a uniform distribution of weights does not map to a uniform set of solutions on the Pareto front (Das & Dennis, 1998), frequently resulting in clustered solutions and large geometric gaps.

In this work, we propose *Sequential Targeting via Adaptive Gap-Filling $\varepsilon$-Constraint Bayesian Optimization (STAGE-BO)* to efficiently generate uniform Pareto optimal solutions without the biases or costs of hypervolume computation. We build upon the key insight of the $\varepsilon$-constraint method (Haimes, 1971; Chankong & Haimes, 2008; Branke, 2008): any Pareto-optimal solution can be recovered by optimizing one objective while constraining the others (Mavrotas, 2009). The practical challenge lies in selecting constraint thresholds $\varepsilon$ that leads to uniform coverage of the Pareto front. Our method solves this issue by identifying the largest gap between the surrogate Pareto front and the observations, which are then set as constraints in the $\varepsilon$-constraint method. This way, we transform the global MOBO problem into a sequence of inequality-constrained sub-problems solved via constrained expected improvement (Schonlau et al., 1998; Gardner et al., 2014).

Our specific contributions are as follows:

- We propose STAGE-BO: an adaptive gap-filling $\varepsilon$-constraint method that selects constraints based on the geometry of the surrogate Pareto front (measured by *fill distance*), thereby guaranteeing uniform coverage without hypervolume computation.

- We introduce a general framework that seamlessly handles standard MOO, constrained MOO, and preference-aware MOO without requiring structural modifications.

- We demonstrate through extensive experiments that STAGE-BO achieves superior Pareto coverage and competitive hypervolume performance compared to state-of-the-art baselines on both synthetic benchmarks and real-world tasks.

## 2. Background

### 2.1. Bayesian Optimization

Bayesian Optimization (BO) (Garnett, 2023) is a sequential design strategy for the global optimization of black-box functions that are expensive to evaluate. Formally, we seek to find a global maximizer $\mathbf{x}^*$ of an objective function: $f : \mathcal{X} \rightarrow \mathbb{R}$ over a bounded domain $\mathcal{X} \subset \mathbb{R}^d$: $\mathbf{x}^* = \arg\max_{\mathbf{x} \in \mathcal{X}} f(\mathbf{x})$. The BO framework rests on two principal components: a probabilistic surrogate model and an acquisition function. A Gaussian Process (GP) (Williams & Rasmussen, 2006) is typically employed as a prior distribution over $f(\mathbf{x})$. The GP is fully specified by a mean function and a covariance kernel, denoted as $f(\mathbf{x}) \sim \mathcal{GP}(m(\mathbf{x}), k(\mathbf{x}, \mathbf{x}'))$. Given a dataset of observations $\mathcal{D}_t = \{(\mathbf{x}_i, \mathbf{y}_i)\}_{i=1}^t$, an acquisition function $\alpha(\mathbf{x}|\mathcal{D}_t)$ is maximized to sample the next query: $\mathbf{x}_{t+1} = \arg\max_{\mathbf{x} \in \mathcal{X}} \alpha(\mathbf{x}|\mathcal{D}_t)$. The objective $f$ is then evaluated at $\mathbf{x}_{t+1}$, the dataset is updated and the posterior is recomputed. The common acquisition functions are EI (Močkus, 1974), UCB (Srinivas et al., 2010) and TS (Thompson, 1933).

### 2.2. Multi-Objective Optimization

A multi-objective optimization (MOO) problem has a vector-valued objective function $F : \mathcal{X} \rightarrow \mathcal{Y}$ with $F = (f_1, \ldots, f_m)$, where $\mathcal{X} \in \mathbb{R}^d$ is a $d$-dimensional input space, and $\mathcal{Y} \in \mathbb{R}^m$ is an $m$-dimensional output space ($m > 1$). Without loss of generality, we assume the problem is to maximize all objectives of $F$: $\max_{\mathbf{x} \in \mathcal{X}} F(\mathbf{x}) = [f_1(\mathbf{x}), \ldots, f_m(\mathbf{x})]$. In MOO, the goal is to identify the set of *Pareto optimal solutions*, all of which are mathematically equivalent when no preference information is specified.

**Pareto Optimality** For a pair $(\mathbf{x}, \mathbf{x}')$, we say "$\mathbf{x}$ weakly dominates $\mathbf{x}'$" if $F(\mathbf{x})$ is no worse than $F(\mathbf{x}')$ in all objectives, i.e. $f_i(\mathbf{x}) \geq f_i(\mathbf{x}')$ for all $i \in \{1, \ldots, m\}$. If at least one of the inequalities is strict, we say "$\mathbf{x}$ dominates $\mathbf{x}'$". If $\mathbf{x}$ is not (weakly) dominated by any other $\mathbf{x}'$, $\mathbf{x}$ is called (weakly) Pareto-optimal. *(Weak) Pareto front $\mathcal{P}_f$* is a set of (weakly) Pareto optimal solutions, and the corresponding set of Pareto optimal inputs is called the *Pareto set $\mathcal{P}_s$*.

Multi-Objective Bayesian Optimization (MOBO) extends BO to optimize expensive black-box, vector-valued objective functions $F$. Given a maximization problem, the goal is to identify the Pareto set $\mathcal{P}_s$ and the corresponding Pareto front $\mathcal{P}_f$, using a minimal number of function evaluations. MOBO frameworks typically employ independent GPs to model each objective function $f_i$ (Bradford et al., 2018; Paria et al., 2020; Belakaria et al., 2020; Daulton et al., 2020). An acquisition function is optimized to select the next query. See the detailed discussion in Section 3.

Two commonly used performance metrics in MOO are hy-

pervolume and inverted generational distance. *Hypervolume* (HV) (Zitzler & Thiele, 1998) is defined as the $m$-dimensional Lebesgue measure $\lambda_m$ of the space dominated by solutions and bounded by the reference point $\mathbf{r}$:

$$\text{HV}(\mathbf{Y}, \mathbf{r}) = \lambda_m(\cup_{\mathbf{y} \in \mathbf{Y}}[\mathbf{r}, \mathbf{y}]), \tag{1}$$

where $[\mathbf{r}, \mathbf{y}]$ denotes the hyperrectangle bounded by the reference point $\mathbf{r}$ and solutions $\mathbf{y} \in \mathbf{Y}$.

*Inverted generational distance* (IGD) (Coello Coello & Reyes Sierra, 2004) assesses both the convergence and diversity of the approximation by measuring the average distance from the true Pareto front $\mathcal{P}_f$ to the nearest solution in the observed set $\mathbf{Y}$:

$$\text{IGD}(\mathbf{Y}, \mathcal{P}_f) = \frac{1}{|\mathcal{P}_f|}\left(\sum_{\mathbf{y}' \in \mathcal{P}_f} \min_{\mathbf{y} \in \mathbf{Y}} \|\mathbf{y} - \mathbf{y}'\|\right). \tag{2}$$

Low IGD values indicate that the front approximation is both close to the true front and covers it uniformly.

### 2.3. Constrained Multi-Objective Optimization

In practical engineering and scientific applications, valid solutions often satisfy safety or physical constraints alongside objective trade-offs (Fromer & Coley, 2023; Gardner et al., 2019). The constrained multi-objective optimization problem is defined as:

$$\begin{aligned} \max_{\mathbf{x} \in \mathcal{X}} F(\mathbf{x}) &= [f_1(\mathbf{x}), \ldots, f_m(\mathbf{x})] \\ s.t.\ G(\mathbf{x}) &= [g_1(\mathbf{x}), \ldots, g_q(\mathbf{x})] \geq 0, \end{aligned} \tag{3}$$

where $G(\mathbf{x}) : \mathcal{X} \to \mathcal{C} \in \mathbb{R}^q$ is the $q$ constraints functions. Both $F(\mathbf{x})$ and $G(\mathbf{x})$ are unknown. Consequently, the search space is restricted to the feasible region given by

$$\mathcal{Q} = \{\mathbf{x}|\mathbf{x} \in \mathcal{X}, g_l(\mathbf{x}) \geq 0, \forall l \in [q]\}. \tag{4}$$

The goal is to identify the Pareto front $\mathcal{P}_f$ and Pareto set $\mathcal{P}_s$ strictly within $\mathcal{Q}$. This setting introduces complexity, as the optimizer must simultaneously learn the boundaries of the feasible region and maximize the objectives.

A standard approach to constrained MOBO is model each objective $f_i(\mathbf{x})$ and constraint $g_j(\mathbf{x})$ using independent GPs. The search for feasible Pareto front is guided by modifying the acquisition function to account for constraint satisfaction, such as cEHVI (Abdolshah et al., 2018), qPOTS (Renganathan & Carlson, 2025), COMBOO (Li et al., 2025).

### 2.4. Preference-Aware Multi-Objective Optimization

While standard MOO aims to approximate the entire Pareto front, in many decision-making scenarios, the whole front is computationally expensive to recover (Paria et al., 2020;

Chen et al., 2024). Instead, decision-makers often possess prior knowledge regarding acceptable trade-offs. This motivates the preference-aware setting, where the goal is to concentrate evaluations solely on a user-defined Region of Interest (ROI). Following Paria et al. (2020); Hakanen & Knowles (2017), we adopt the bounding box formulation, where preferences are expressed as thresholds. The ROI is defined as $\mathcal{B} = \{\mathbf{y} \in \mathbb{R}^m | a_i \leq \mathbf{y} \leq b_i, \forall i = 1, \ldots, m\}$. The optimization goal is to recover the Pareto optimal solutions satisfying $F(\mathbf{x}) \in \mathcal{B}$.

## 3. Related Work

**Multi-Objective Bayesian Optimization** MOBO methods largely fall into three categories: scalarization, hypervolume (HV) maximization, and information-theoretic approaches. Scalarization methods (ParEGO (Knowles, 2006) and TS-TCH (Paria et al., 2020)) decompose the problem into single-objective subtasks using random weights, which often fail to cover fronts uniformly. HV-based methods (EHVI (Emmerich & Klinkenberg, 2008), qEHVI (Daulton et al., 2020), and TSEMO (Bradford et al., 2018)), prioritize maximizing the dominated volume. As discussed in Section 1, they suffer from the intrinsic bias and high computational costs that scale exponentially with the number of objectives. Information-theoretic methods (PESMO (Hernández-Lobato et al., 2016), MESMO (Belakaria et al., 2019), PFES (Suzuki et al., 2020), and JESMO (Tu et al., 2022)) maximize information gain about the Pareto front but often require heavy approximations to compute.

Recent work attempts to explicitly enforce coverage but typically retains bottlenecks. DGEMO (Konakovic Lukovic et al., 2020) guides the search toward diverse regions but still employs HV improvement for the final selection. Moreover, its reliance on data structures limits the scalability beyond three objectives. PDBO (Ahmadianshalchi et al., 2024) employs a bandit strategy with determinantal point processes to select diverse batches. MOBO-OSD (Ngo et al., 2025) decomposes the problem via orthogonal search directions but still relies on HV maximization for selection. qPOTS (Renganathan & Carlson, 2025) combines Thompson Sampling with a maximin strategy; however, it calculates diversity in the input space, which does not guarantee output space diversity. Unlike these approaches, our method ensures output-space diversity without calculating HV.

**Constrained Multi-Objective Bayesian Optimization** Research explicitly targeting constrained MOBO remains relatively sparse. Standard frameworks, such as the widely used BoTorch (Balandat et al., 2020) implementations of qEHVI (Daulton et al., 2020; 2021), typically handle constraints by weighting the primary acquisition value by the probability of feasibility, adopting the strategy originally

proposed by Gelbart et al. (2014). Within the information-theoretic paradigm, Hernández-Lobato et al. (2016) extended Predictive Entropy Search (PES) to constrained settings, explicitly balancing constraint learning with Pareto frontier discovery. To mitigate the high computational cost of PES, Fernández-Sánchez et al. (2023) subsequently proposed a formulation based on Max-value Entropy Search (MES) (Wang & Jegelka, 2017). However, these entropy-based methods rely on heavy approximations to maintain tractability, which can degrade performance in complex landscapes. Most recently, qPOTS (Renganathan & Carlson, 2025) can also handle constraints naturally. COMBOO (Li et al., 2025) introduced a scalarization-based approach that combines random weights with optimistic feasibility assessments (UCB) to identify the constrained Pareto front.

**Preference-Aware Multi-Objective Bayesian Optimization** The integration of Decision Maker preferences has been stuided in MOO. Early approaches (Abdolshah et al., 2019) encoded these preferences via objective importance rankings. However, a more prevalent formulation defines the preference structure as a specific Region of Interest (ROI) in the objective space, typically bounded by reference vectors or hyper-rectangles (Hakanen & Knowles, 2017; Paria et al., 2020; Palar et al., 2018; He et al., 2020). Paria et al. (2020) systematically generalize the random scalarization technique, allowing different scalarization techniques, e.g., weighted sum and Tchebyshev (Nakayama et al., 2009), as well as different acquisition functions, e.g., TS (Thompson, 1933), UCB (Srinivas et al., 2010) to concentrate candidate generation specifically within the user's preferred region.

## 4. Methodology

Our key motivation is the observation that optimizing hypervolume may lead to non-uniformly distributed Pareto optimal solutions and is computationally expensive. This work aims to generate a set of Pareto optimal solutions that uniformly cover $\mathcal{P}_f$. We exploit the property that the $\varepsilon$-constraint method can find any Pareto-optimal point with the appropriate constraints on the objectives. Thus, the challenge is finding the constraints such that the queries found by the $\varepsilon$-constraint method lead to uniform coverage. Our method, called STAGE-BO, tackles this by identifying the largest under-explored region (gap) in the objective space.

To measure and, ultimately, optimize for uniform coverage, we adopt the *Fill Distance* (FD) metric defined in Zhang et al. (2024). For $\mathcal{D}_t = \{\mathbf{X}_t, \mathbf{Y}_t\}$,

**Definition 4.1.** $\mathrm{FD}(\mathbf{Y}_t) = \max_{\mathbf{y} \in \mathcal{P}_f} \min_{\mathbf{y}' \in \mathbf{Y}_t} \|\mathbf{y} - \mathbf{y}'\|$,

where $\| \cdot \|$ denotes the Euclidean distance between two points. Zhang et al. (2024) establish the relationship between FD and IGD and prove that the optimal FD configuration sets an upper bound for the IGD value.

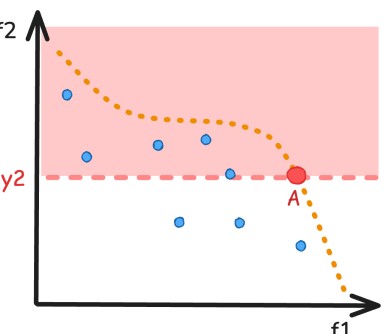

*Figure 2.* Illustration of the STAGE-BO algorithm. The blue dots represent observations. The dashed orange curve depicts the sampled Pareto front approximation ($\widetilde{\mathcal{P}}_f^t$), generated via Thompson sampling and NSGA-II. The red point $A = (y_1, y_2)$ is identified on $\widetilde{\mathcal{P}}_f^t$ as having the maxmin distance to the existing blue observations. Assuming a schedule where $f_1$ is targeted for optimization at this step, a constraint is established on the remaining objective using the objective value of $A$ ($f_2 \geq y_2$). The resulting valid search space for maximizing $f_1$ is indicated by the light red shaded region.

**Theorem 4.2** (Zhang et al. (2024))**.** *Assume the goal is to find a set of Pareto optimal solutions, i.e., $\mathbf{Y} \subset \mathcal{P}_f$ to optimize either the FD or IGD indicator: $\min_{\mathbf{Y} \subset \mathcal{P}_f} FD(\mathbf{Y})$ or $\min_{\mathbf{Y} \subset \mathcal{P}_f} IGD(\mathbf{Y})$ to reach a diverse distribution. Let the optimal sets be $\mathbf{Y}^{FD}$ and $\mathbf{Y}^{IGD}$ respectively. Then*

$$IGD(\mathbf{Y}^{IGD}) \leq IGD(\mathbf{Y}^{FD}) \leq FD(\mathbf{Y}^{FD}). \quad (5)$$

Since the optimal IGD configuration does not similarly bound FD, we focus on minimizing FD in this paper.

### 4.1. Proposed Method: STAGE-BO

STAGE-BO consists of four main steps: 1) Sample the surrogate objective functions; 2) Identify the Pareto front point with the largest uncovered gap, quantified by FD; 3) Translate the target's coordinates into constraints, formulating the MOO problem into a sequence of inequality-constrained subproblems; 4) Solve the resulting constrained optimization problem. We present the details below.

FD identifies locations that are not yet covered by observations. Since the true Pareto front $\mathcal{P}_f$ is not known, we resort to sampling from the posterior.

**Target Identification** We employ Thompson sampling (Wilson et al., 2020) to approximate the objectives. Specifically, we draw a spectral sample path

$$\tilde{F}^t(\mathbf{x}) = [\tilde{f}_1^t(\mathbf{x}), \ldots, \tilde{f}_m^t(\mathbf{x})], \ \tilde{f}_i^t(\cdot) \sim \mathcal{GP}(f_i | \mathcal{D}_t) \quad (6)$$

from the posterior distribution of the objective functions. We then compute the *sampled Pareto front $\widetilde{\mathcal{P}}_f^t$* by maximizing this sampled trajectory:

$$\widetilde{\mathcal{P}}_f^t = \max_{\mathbf{x} \in \mathcal{X}} [\tilde{f}_1(\mathbf{x}), \ldots, \tilde{f}_m(\mathbf{x})]. \quad (7)$$

This cheap MOO problem is solved using evolutionary algorithms such as NSGA-II (Deb et al., 2002), generating a discrete approximation of the front.

Another option is to use the posterior mean as the surrogate path. However, as our ablation study in Appendix F.1 shows, replacing Thompson-sampled path with the posterior mean results in worse performance. This is because the posterior mean is overly greedy and suppresses the uncertainty-driven variability needed for effective exploration.

On the sampled front $\widetilde{\mathcal{P}}_f$, we seek the target $\mathbf{Y}_c$ that approximates the center of the largest under-explored region from our current observations $\mathbf{Y}_t$:

$$\mathbf{Y}_c = \arg \max_{\mathbf{y}' \in \widetilde{\mathcal{P}}_f^t} \min_{\mathbf{y} \in \mathbf{Y}_t} \|\mathbf{y} - \mathbf{y}'\|. \tag{8}$$

**$\varepsilon$-constraint Decomposition**    Having identified the target location $\mathbf{Y}_c$, we require an optimization mechanism to guide the search toward it. To achieve this, we employ the $\varepsilon$-constraint method, which allows us to translate the coordinates of $\mathbf{Y}_c$ directly into search space boundaries. By setting the constraint thresholds based on the coordinates of $\mathbf{Y}_c$, we transform the multi-objective problem into a constrained single-objective subproblem whose unique optimal solution is guaranteed to be Pareto optimal (Branke, 2008).

$$\max_{\mathbf{x} \in \mathcal{X}} \ f_k(\mathbf{x}) + s \sum_j f_j$$
$$\text{subject to } f_j(\mathbf{x}) \geq \varepsilon_j \text{ for all } j = 1, \dots, m, j \neq k. \tag{9}$$

$s$ is set as a small number (e.g., $10^{-3}$) to avoid weakly Pareto optimal points. This ensures that our next query is optimally positioned to fill the identified void. The detailed discussion can be found in Appendix B.

To avoid defining an empty feasible region, if $j$-th objective of the target component $\mathbf{Y}_{c,j}$ exceeds the maximum observed value for $j$-th objective, we clip the constraint to the best observed value:

$$\widehat{\mathbf{Y}}_{c,j} = \begin{cases} \mathbf{Y}_{c,j} & \text{if } \mathbf{Y}_{c,j} < \mathbf{Y}_{t,j} \ \exists t; \\ \max\{\mathbf{Y}_{t,j}\}_t & \text{if } \mathbf{Y}_{c,j} \geq \mathbf{Y}_{t,j} \ \forall t. \end{cases} \tag{10}$$

This clipping rule is designed as a numerical stabilizer: it is triggered precisely when the target threshold exceeds all current observations on that objective, and usually happens in the early stage of the algorithm. An ablation study comparing STAGE-BO with and without clipping in Appendix F.2 shows that on most benchmarks the two variants perform comparably, confirming that clipping acts primarily as a stabilizer. On a subset of benchmarks, clipping leads to measurable improvements, suggesting that the larger feasible region induced by clipping benefits optimization.

To ensure balanced exploration across the objective space, we rotate the objective $f_k$ to be optimized in a round-robin

fashion ($k = t \pmod{m} + 1$). This schedule guarantees that optimization pressure is distributed uniformly across all objectives over time. As demonstrated in Appendix F.4, our framework is robust to the specific strategy used to select the objective for optimization.

**Acquisition Optimization**    This decomposition transforms the original MOBO problem into a generic constrained Bayesian optimization problem. The constraints are placed at $m - 1$ objectives $\{j | j \in (1, \dots, m), \text{ and } j \neq k\}$ with thresholds $\varepsilon_j = \widehat{\mathbf{Y}}_{c,j}$ and $k$-th objective is set to be optimized. We solve this efficiently using the Constrained Expected Improvement (cEI) acquisition function: (Schonlau et al., 1998; Gardner et al., 2014).

$$\mathbf{x}_{t+1} = \arg \max_{\mathbf{x} \in \mathcal{X}} \text{EI}(\mathbf{x}) \times \text{PoF}(\mathbf{x}), \tag{11}$$

where $\text{EI}(\mathbf{x})$ is the standard Expected Improvement of the objective $f_k$, and $\text{PoF}(\mathbf{x})$ is the Probability of Feasibility satisfying the constraints.

$$\text{EI}(\mathbf{x}) = \mathbb{E}[\max(0, f_k(\mathbf{x}) + s \sum_{j \neq k} f_j(\mathbf{x}) - f_k^* - s \sum_{j \neq k} f_j^*], \tag{12}$$

where $f_k^* + s \sum_{j \neq k} f_j^*$ is the best observed point. We assume the constraints are independent GPs.

$$\text{PoF}(\mathbf{x}) = \prod_{j=1,\dots,m, j \neq k} \text{Pr}(f_j(\mathbf{x}) \geq \widehat{\mathbf{Y}}_{c,j}). \tag{13}$$

By solving Equation (11), STAGE-BO selects the sample that optimizes Equation (9), effectively filling the identified gap. We visually show this process in Appendix A.

---

**Algorithm 1** The STAGE-BO Algorithm
---
1: **Input:** Evaluation budget $T$, initial dataset $\mathcal{D}_0$.
2: **while** $t \leq T$ **do**
3:     Fit GP models on $\mathcal{D}_t$.
4:     Sample the posterior GP paths (Equation (6)).
5:     Solve the cheap MOO problems Equation (7) via NSGA-II.
6:     Identify the target point $\mathbf{Y}_c$ on $\widetilde{\mathcal{P}}_f$ with the maxmin distance to evaluations $\mathbf{Y}_t$ in Equation (8).
7:     Select primary objective: $k \leftarrow t \pmod{m} + 1$.
8:     Set the constraints based on Equation (10).
9:     Optimize cEI (Equation (11)) to sample next point $(\mathbf{x}_{t+1}, \mathbf{y}_{t+1})$.
10:    Update the dateset $\mathcal{D}_{t+1} \leftarrow \mathcal{D}_t \cup (\mathbf{x}_{t+1}, \mathbf{y}_{t+1})$.
11: **end while**
12: **Output:** The Pareto set $\mathcal{P}_s$ and Pareto front $\mathcal{P}_f$.
---

## 4.2. Extensions to constrained MOBO

In practice, regulatory or safety concerns often impose additional thresholds $G(\mathbf{x})$ on certain attributes of the experimental outcomes (Fromer & Coley, 2023). STAGE-BO extends naturally to this setting without structural changes.

Since our framework already reduces MOO to a sequence of constrained sub-problems (Equation (9)), incorporating physical constraints is natural: we append the external constraints to the set of algorithmically generated $\varepsilon$-constraints. The optimization problem becomes:

$$\max f_k(\mathbf{x}) + s \sum_j f_j$$

$$\text{subject to } f_j(\mathbf{x}) \geq \varepsilon_j \text{ for all } j = 1, \ldots, m, j \neq k$$
$$g_l(\mathbf{x}) \geq 0 \text{ for all } l = 1, \ldots, q. \quad (14)$$

To ensure the identified target $\mathbf{Y}_c$ is physically reachable, we must also account for feasibility during the target identification phase. After collecting $t$ observations $\mathcal{D}_t = \{\mathbf{X}_t, \mathbf{Y}_t, \mathbf{C}_t)\} = \{(\mathbf{x}_i, \mathbf{y}_i, \mathbf{c}_i)\}_{i=1}^t$, we build GP models separately for each objective and each constraint. We sample paths for both objectives and constraints using Thompson sampling. Besides Equation (6), we also sample

$$\tilde{G}^t(\mathbf{x}) = [\tilde{g}_1^t(\mathbf{x}), \ldots, \tilde{g}_c^t(\mathbf{x})], \; \tilde{g}_i^t(\cdot) \sim \mathcal{GP}(g_i | \mathcal{D}_t). \quad (15)$$

We optimize the following cheap constrained MOO problem to obtain the sampled Pareto front $\widetilde{\mathcal{P}}_f^t$ using NSGA-II:

$$\widetilde{\mathcal{P}}_f^t = \max_{\mathbf{x} \in \mathcal{X}} [\tilde{f}_1(\mathbf{x}), \ldots, \tilde{f}_m(\mathbf{x})],$$

$$\text{subject to } \tilde{g}_l(\mathbf{x}) \geq 0, \; \forall l \in [q] \quad (16)$$

This ensures that the target gap $\mathbf{Y}_c$ computed via Equation (8) lies within the valid feasible region. Finally, the acquisition optimization (Equation (11)) remains unchanged, with PoF($\mathbf{x}$) updated to include the probability of satisfying physical constraints:

$$\text{PoF}(\mathbf{x}) = \prod_{j \neq k} \text{Pr}(f_j(\mathbf{x}) \geq \widehat{\mathbf{Y}}_{c,j}) \prod_{l=1}^q \text{Pr}(g_l(\mathbf{x}) \geq 0). \quad (17)$$

## 4.3. Extensions to MOBO with preferences

In many decision-making scenarios, exploring the entire Pareto front is unnecessary (Paria et al., 2020; Hakanen & Knowles, 2017). Instead, domain experts often define a Region of Interest (ROI), typically specified as a hyper-rectangle bounded by $f_i \in [a_i, b_i], i = 1, \ldots, m$. Our framework naturally incorporates these preferences by integrating the boundaries directly into the constraint set.

In preference-based setting, the user-defined ROI is often misaligned with the true feasible space: it may be overly

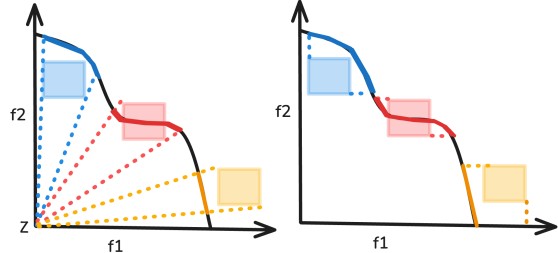

*Figure 3.* **Comparison of preference handling strategies. Left:** Preference scalarization methods (Paria et al., 2020) map preferred regions (shaded boxes) to preference weights dependent on a reference point $z$. **Right:** Our geometric approach operates without a reference point, directly targeting Pareto-optimal solutions that satisfy either the lower or upper bounds of the specified regions.

ambitious (completely beyond the true front) or overly conservative (dominated). To address this issue, we do not treat the ROI as a rigid binary target. Instead, we interpret the ROI as defining two distinct anchor constraints: the lower bounds (minimum acceptable criteria) and the upper bounds (ideal limits). We formulate the optimization problem to seek Pareto-optimal solutions that satisfy either of these boundary sets (Figure 3):

$$\max f_k(\mathbf{x}) + s \sum_j f_j \quad (18)$$

$$\text{subject to } f_j(\mathbf{x}) \geq \varepsilon_j \text{ for all } j = 1, \ldots, m, j \neq k$$
$$\begin{cases} (f_j(\mathbf{x}) \geq a_j \text{ for all } j = 1, \ldots, m, j \neq k) \\ \quad\quad\quad \text{or} \\ (f_j(\mathbf{x}) \leq b_j \text{ for all } j = 1, \ldots, m, j \neq k). \end{cases}$$

This formulation ensures robustness: if the aspirational upper bounds are unreachable, the solver anchors to the lower bounds to recover the best valid trade-offs. Conversely, if the lower bounds are trivially satisfied, the upper bounds drive the search toward superior regions.

To focus the search within the preferred region, the preference constraints are incorporated in the target identification. After sampling the path in Equation (6), we solve the following cheap MOO problem with NSGA-II to obtain the sampled Pareto front in the preferred region $\widetilde{\mathcal{P}}_f^t$

$$\widetilde{\mathcal{P}}_f^t = \max_{\mathbf{x} \in \mathcal{X}} [\tilde{f}_1(\mathbf{x}), \ldots, \tilde{f}_m(\mathbf{x})], \quad (19)$$

$$\text{subject to } \begin{cases} (\tilde{f}_j(\mathbf{x}) \geq a_j \text{ for all } j = 1, \ldots, m, j \neq k). \\ \quad\quad\quad \text{or} \\ (\tilde{f}_j(\mathbf{x}) \leq b_j \text{ for all } j = 1, \ldots, m, j \neq k). \end{cases}$$

Crucially, once the target coordinate $\mathbf{Y}_c$ is identified within the preferred region, the preference information is implicitly encoded into the gap-filling $\varepsilon$-constraints ($\varepsilon_j = \widehat{\mathbf{Y}}_{c,j}$). Therefore, the acquisition function remains identical to the standard version in Equation (11).

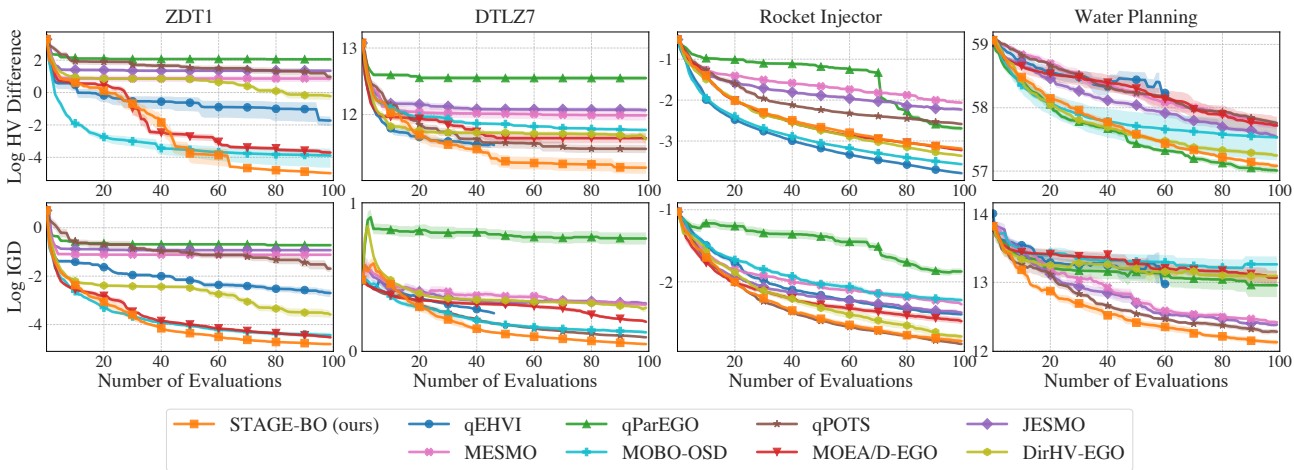

*Figure 4.* Comparison of our method with state-of-the-art baselines on two synthetic and two real-world benchmark **MOO problems**. The first row reports hypervolume, and the second row reports IGD. Overall, our method achieves comparable or superior hypervolume relative to the baselines and consistently outperforms them in terms of IGD.

## 5. Experiments

We empirically evaluate our proposed method against the state-of-the-art methods on an extensive set of synthetic and real-world benchmark problems under different tasks. Lastly, we present results on hyperparameter optimization for privacy-preserving machine learning, demonstrating the practical effectiveness of STAGE-BO.

### 5.1. Unconstrained MOBO

**Settings and Baselines.** We evaluate STAGE-BO against a comprehensive set of baselines: qEHVI (Daulton et al., 2020), qParEGO (Knowles, 2006; Daulton et al., 2020), JESMO (Tu et al., 2022), MESMO (Belakaria et al., 2019), qPOTS (Renganathan & Carlson, 2025), MOBO-OSD (Ngo et al., 2025), MOEA/D-EGO (Zhang et al., 2009) and DirHV-EGO (Zhao & Zhang, 2023). $q = 1$. Detailed implementations of our method and the baselines can be found in Appendix G.

We conduct experiments on six benchmark problems. The number of objectives ranges from two to six, which is common in the MOBO literature. For synthetic benchmark problems, we consider ZDT1 ($d = 10, m = 2$), ZDT2 ($d = 8, m = 2$) and DTLZ7 ($d = 6, m = 5$) with a discontinuous Pareto front. For the real-world benchmark problems, we consider problems from the problem suite (Tanabe & Ishibuchi, 2020): Coil compression spring design ($d = 3, m = 2$), Rocket injector design ($d = 4, m = 3$) and Water resource planning ($d = 3, m = 6$). These problems are widely used in the MOBO literature (Daulton et al., 2023; Ngo et al., 2025; Renganathan & Carlson, 2025). Details of benchmark problems can be found in Appendix H.

For the evaluation metrics, we compute the difference be-

tween the HV of the observed Pareto front and the maximum HV. We also report IGD, which indicates convergence and diversity. We report the mean and the standard error across 10 independent runs.

**Results.** Figure 4 summarizes the performance of STAGE-BO and all baselines. Note that qEHVI is evaluated for a limited number of iterations on DTLZ7 and the water planning design problem due to the prohibitively high computational cost when the number of objectives satisfies $m \geq 4$. Our method consistently outperforms baselines with respect to IGD, indicating faster convergence toward the Pareto front and improved solution diversity. Although our method does not explicitly optimize HV, it consistently performs comparable to the best baselines, thanks to diverse coverage of the Pareto front as suggested by the low IGD.

Additional performance metrics, IGD+ and fill distance, are provided in Appendix D.1. The full results for the ZDT2 benchmark and the Coil Compression Spring design problem are detailed in Appendix D.2 due to space limit. Furthermore, we provide a theoretical time complexity analysis for STAGE-BO, along with a runtime comparison between STAGE-BO and the baselines in Appendix E, demonstrating that STAGE-BO maintains high efficiency in both low- and high-dimensional objective spaces.

We also conduct ablation studies in Appendix F to validate the core components of our framework—our fill-distance-based constraints, the cEI acquisition function and the Thompson-sampled path. Note that STAGE-BO is not specific to cEI. Other constrained BO acquisition functions could also be used. We discuss this design choice in Appendix C. The results also verify the clipping rule (Equation 10) act as a numerical stabilizer. Furthermore, we demon-

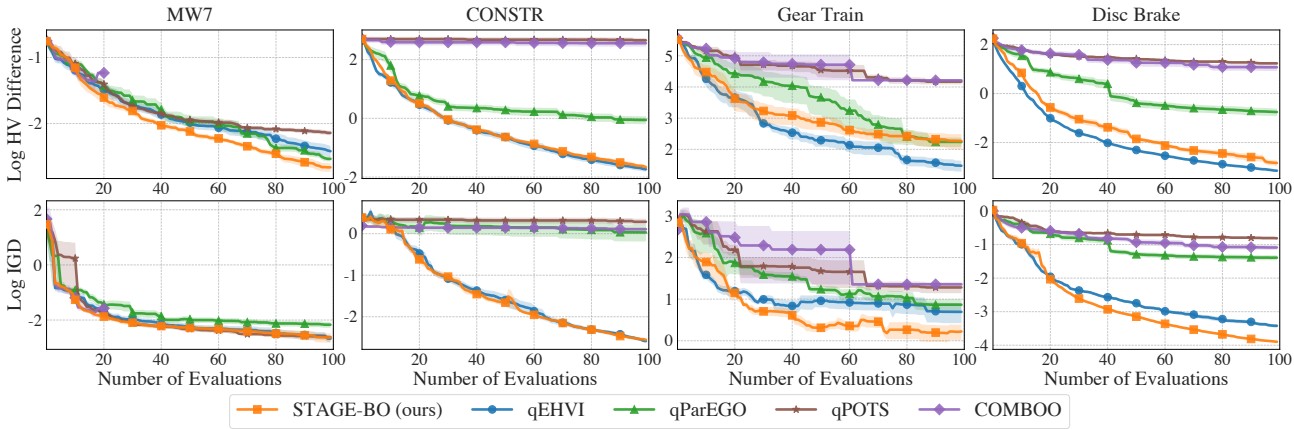

*Figure 5.* Comparison of our method with state-of-the-art baselines on one synthetic and three real-world benchmark **constrained MOO problems**. The first row reports hypervolume, and the second row reports IGD. Overall, our method achieves comparable or superior hypervolume relative to the baselines and consistently outperforms them in terms of IGD.

strate that STAGE-BO is insensitive to the specific strategy used for selecting the primary optimization objective.

### 5.2. Constrained MOBO

**Settings and Baselines.** We evaluate STAGE-BO against a set of baselines: qEHVI, qParEGO, qPOTS, COMBOO (Li et al., 2025). Detailed implementations of our method and the baselines can be found in Appendix G.

We conduct experiments on four constrained MOO benchmark problems. For synthetic benchmark problem, we consider MW7 ($d = 4, m = 2, c = 2$). For the real-world benchmark problems, we consider two problems from the problem suite: (Tanabe & Ishibuchi, 2020): Disc brake design ($d = 4, m = 2, c = 4$), Gear train design ($d = 4, m = 2, c = 1$), and one problem from Garrido-Merchán & Hernández-Lobato (2020): CONSTR ($d = 2, m = 2, c = 2$). Details of the benchmark problems can be found in Appendix H.

**Results.** Figure 5 shows the performance of all methods. Note that COMBOO is evaluated with a limited number of function evaluations on MW7, as it terminates early and returns the observed solutions when the UCB-based estimates of all constraints indicate infeasibility. STAGE-BO consistently outperforms state-of-the-art methods in terms of IGD, demonstrating both faster convergence and superior solution diversity. While our framework does not explicitly optimize for HV, its ability to ensure uniform coverage of the Pareto front inherently leads to HV performance that is competitive with, or superior to baselines. The results on Gear Train and Disc Brake indicate a trade-off between IGD and HV, as no single method achieves the best performance in both metrics. More metrics including the feasible evaluation ratio, IGD+, and fill distance are reported in Appendix D.1

### 5.3. Preference-Aware MOBO

**Settings and Baselines.** We evaluate STAGE-BO against a set of baselines: qEHVI, qParEGO, qPOTS, TSTCH (Paria et al., 2020). Detailed implementations of our method and the baselines can be found in Appendix G.

We conduct experiments on two synthetic benchmark problems: ZDT3 ($d = 2, m = 2$) with the preferred region in ($[-0.7, -0.6], [-0.2, -0.4]$), DTLZ2 ($d = 6, m = 5$) with the preferred region in $[-0.4, -0.4, -0.4, -0.4, -0.4]$, $[-0.2, -0.2, -0.2, -0.2, -0.2]$), and two real-world benchmark problems: VehicleSafety Problem ($d = 5, m = 3$) with the region in ($[-1680, -7, -0.5], [-1675, -6, -0.3]$), CarSideImpact ($d = 7, m = 4$) with the region in ($[-20, -4.5, -10, -7], [-15, -4, -5, -6]$). Details of the benchmark problems can be found in Appendix H.

**Results.** Figure 6 shows the performance of all methods. Our method consistently performs better than competing approaches in terms of HV and IGD, indicating faster convergence toward the preferred region and improved solution diversity. More metrics, IGD+ and fill distance, are detailed in Appendix D.1. We also vary the bounds of ROI in Figures 13 to 16 to demonstrate that STAGE-BO is agnostic to the location of preferred regions.

### 5.4. Real-world application

We additionally evaluate STAGE-BO on a multi-objective hyperparameter optimization task for privacy-preserving machine learning. Differential privacy (Dwork et al., 2006) provides a formal guarantee that limits how much information about any individual training example can be inferred from the learned model, but stronger privacy protection typically requires injecting more noise during training, which can reduce predictive accuracy. This creates a natural

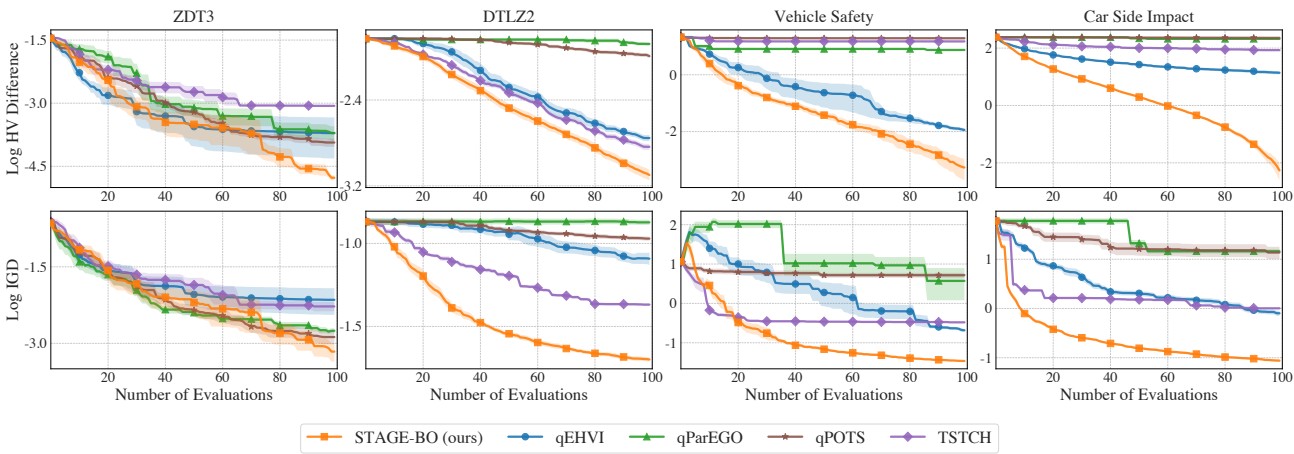

*Figure 6.* Comparison of our method with state-of-the-art baselines on two synthetic and two real-world benchmark MOO problems with preferred regions. The first row reports hypervolume, and the second row reports IGD. Overall, our method achieves superior hypervolume relative to the baselines and consistently outperforms them in terms of IGD.

trade-off between privacy and utility, making the problem well suited to multi-objective optimization. Specifically, we train a logistic regression model with differentially private stochastic gradient descent (DP-SGD) (Abadi et al., 2016) on the Dutch dataset (Van der Laan, 2000), and tune five hyperparameters controlling the training procedure. The hyperparameter ranges are listed in Table 1.

*Table 1.* Hyperparameter search ranges.

| Hyperparameter | Range |
| --- | --- |
| Batch size | $[8, 512]$ |
| Learning rate | $[5e-4, 5e-2]$ |
| Clipping threshold | $[0.1, 4]$ |
| Epochs | $[1, 64]$ |
| Noise level | $[1, 2560]$ |

We consider two competing objectives: model utility and privacy. Because the true Pareto front is unknown in this real-world setting, we evaluate methods using the HV of the observed non-dominated solutions. As shown in Figure 7, STAGE-BO achieves the strongest overall HV performance, demonstrating that the proposed framework remains effective on a practically relevant privacy–utility trade-off beyond the synthetic and engineering benchmarks.

## 6. Conclusion

We have proposed STAGE-BO that bypasses the prohibitive computational costs of hypervolume-based methods by targeting the Pareto front through explicit geometric gap-filling. By reformulating objective trade-offs as adaptive $\varepsilon$-constraints and utilizing cEI, our method provides a unified approach for recovering the global Pareto front, satisfying

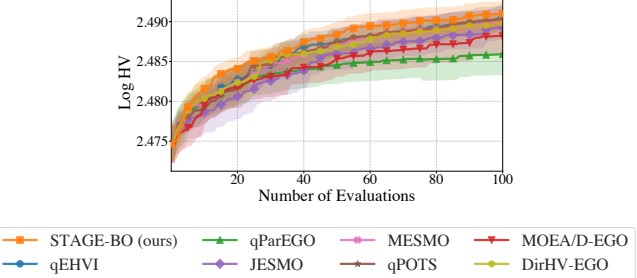

*Figure 7.* Results on a real-world multi-objective hyperparameter optimization task for privacy-preserving machine learning. Since the true Pareto front is unknown, performance is measured by the HV of the observed non-dominated solutions. STAGE-BO achieves the highest HV throughout the optimization process.

physical constraints, or focusing on specific regions of interest. The flexibility and scalability of STAGE-BO make it a robust tool for high-stakes optimization in design and experimental sciences, where balancing diversity, efficiency, and expert preferences is paramount. For example, STAGE-BO could be potentially useful for future interactive MOO methods, where the uses' preferences can be encoded more flexibly than weights (Chen et al., 2025; Yang et al., 2025).

**Limitations** First, STAGE-BO depends on the quality of the sampled Pareto front; if this approximation is poor, target identification may be inaccurate. In our current implementation, we use NSGA-II as the inner solver, which is effective for the low-to-moderate ($m \leq 6$) objective settings considered in this paper. For higher-dimensional many-objective problems, replacing NSGA-II with NSGA-III (Deb & Jain, 2013) may be beneficial. Second, our current gap detection relies on observed point positions, which may be sensitive to measurement noise. Incorporating noise-robust geometric estimates remains a promising direction for future research.

## Impact Statement

This paper presents work whose goal is to advance the field of Machine Learning. There are many potential societal consequences of our work, none which we feel must be specifically highlighted here.

## Acknowledgments

This work was supported by the Research Council of Finland Flagship programme: Finnish Center for Artificial Intelligence FCAI and UKRI Turing AI World-Leading Researcher Fellowship, EP/W002973/1. This work has been performed using resources provided by the Aalto Science-IT Project from Computer Science IT, the CSC – IT Center for Science, Finland and the Finnish Computing Competence Infrastructure (FCCI).

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

## A. Illustration of STAGE-BO

We visually show this process of STAGE-BO in Figure 8. Left and middle panels show independent GP posteriors for the two objectives. A Thompson sample is drawn from the GP posterior for each objective. Optimizing the sampled objectives with NSGA-II yields a sampled Pareto front (green points) in the objective space, shown in the right panel together with the true Pareto front (red points) and the currently observed objective values (black points). STAGE-BO then identifies the target point (purple cross) on the sampled Pareto front by maximizing its minimum Euclidean distance to the observed front, i.e., the largest uncovered geometric gap. This target is converted into an adaptive ε-constraint subproblem, illustrated here by the horizontal constraint line (assume we optimize $f_1$ while constraining $f_2$ at this step). Constrained expected improvement (cEI) is then used to generate the next evaluation (purple star).

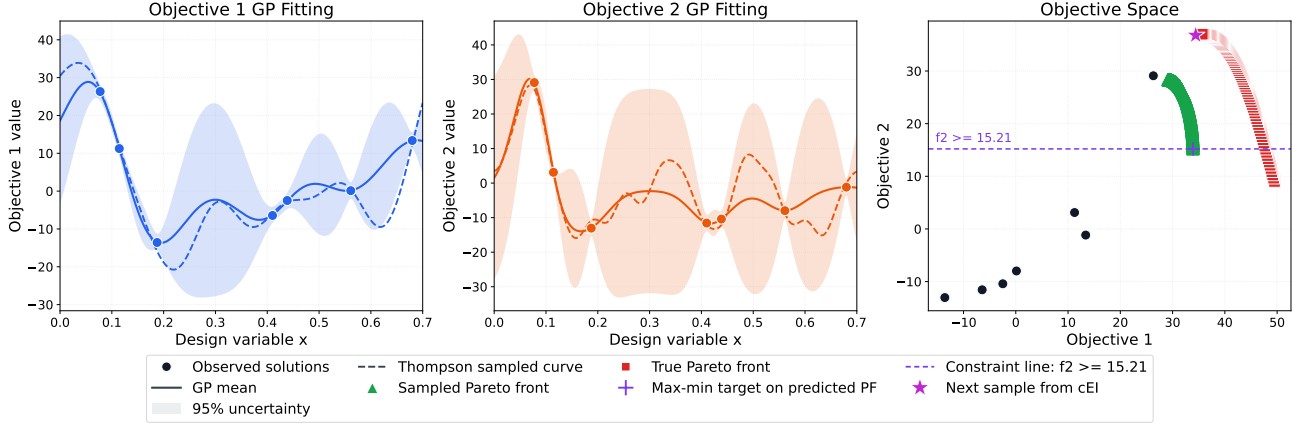

*Figure 8.* Illustration of STAGE-BO on a two-objective problem.

## B. ε-Constraint Multi-Objective Optimization

The ε-constraint method (Haimes, 1971; Chankong & Haimes, 2008) is a classic multi-objective optimization strategy that transforms the original problem into a sequence of single-objective subproblems by treating all but one objective as constraints. While the solution to a standard ε-constraint problem is guaranteed to be at least weakly Pareto optimal (Branke, 2008), the augmented ε-constraint method (Mavrotas, 2009)—Equation (9)—introduces a small slack term to the primary objective to ensure the discovery of strictly Pareto-optimal solutions.

The effectiveness of this approach depends heavily on the placement of the constraint thresholds. To overcome the computational inefficiencies and poor coverage associated with fixed grids, adaptive ε-constraint methods (Laumanns et al., 2006; Liu & Bi, 2021; Fan et al., 2016) have been proposed to iteratively refine these thresholds based on the distribution of previously discovered solutions. While various adaptive ε-constraint schemes exist, they typically rely on systematic grid refinement (Laumanns et al., 2006) or population-based feasibility ratios (Fan et al., 2016) to adjust thresholds. These heuristics are designed for settings with large evaluation budgets, such as evolutionary algorithms, and are not directly applicable to Bayesian Optimization where evaluations are prohibitively expensive. In contrast, our method leverages the posterior GP belief to identify the largest geometric voids in the objective space via fill distance minimization. By placing the ε-constraints specifically at these maxmin coordinates, we transform the ε-constraint method from a passive solver into a proactive, targeted acquisition strategy that ensures global diversity with a minimal number of function evaluations.

## C. Choice of Constrained Solver

By reformulating the MOO problem into a sequence of constrained subproblems, our framework provides the flexibility to utilize more advanced constrained Bayesian optimization methods. In this work, we employ Constrained Expected Improvement (cEI) due to its robustness and mathematical simplicity in handling black-box constraints.

We explicitly distinguish our approach from Trust Region (TR) methods, such as SCBO (Eriksson & Poloczek, 2021). While TR methods are highly effective for optimization with stationary physical constraints, they are fundamentally ill-suited for our framework. In STAGE-BO, the geometric ε-constraints are dynamic targets that shift at every iteration to target the

largest under-explored voids. These shifting feasible regions would frequently invalidate the internal state and local modeling of a trust region, effectively forcing it to restart and thereby negating its primary convergence benefits. Consequently, global acquisition functions like cEI are more appropriate for the moving-target nature of our adaptive decomposition.

## D. Additional Experiments Results

### D.1. Additional Evaluation Metrics

Besides hypervolume and IGD, here we present more evaluation metrics.

**IGD+ (Ishibuchi et al., 2015)**

$$\textbf{IGD+}(\mathbf{Y}_t, \mathcal{P}_f) = \frac{1}{|\mathcal{P}_f|}(\sum_{y \in \mathcal{P}_f} \min_{y' \in \mathbf{Y}_t} d^+(y, y')), \tag{20}$$

where $d^+(y, y') = \sqrt{\sum_{i=1}^m \max(y - y_i, 0)^2}$. IGD+ is weakly Pareto-compliant (Ishibuchi et al., 2015).

**Fill distance (Zhang et al., 2024)**    The definition of fill distance can be found in Theorem 4.1. It measures the maxmin distance between the true Pareto front and the observations.

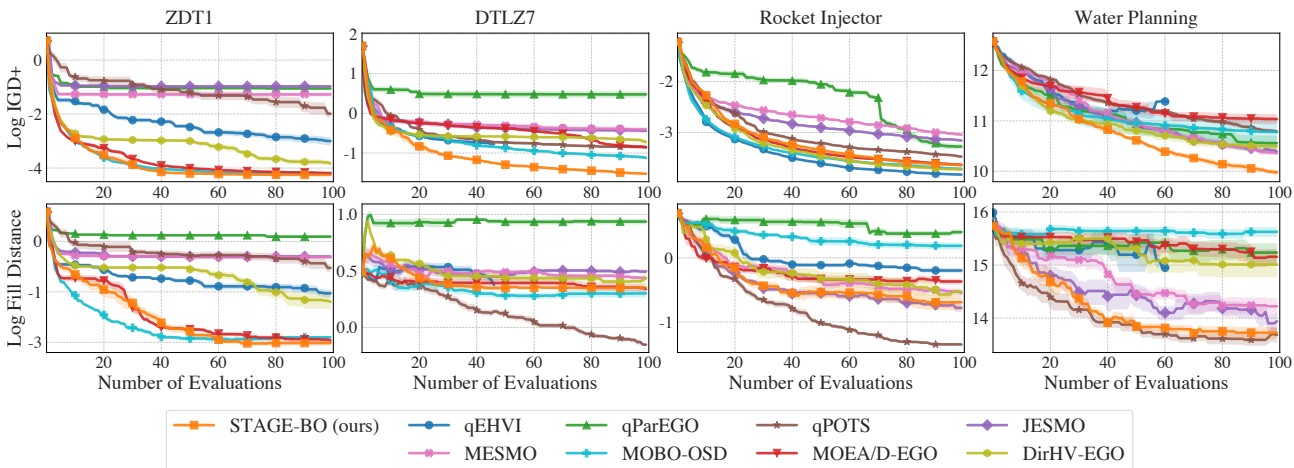

*Figure 9.* Comparison of our method with state-of-the-art baselines on two synthetic and two real-world benchmark **MOO problems**. The first row reports IGD+, and the second row reports fill distance. Overall, our method achieves comparable or superior performance.

Figure 9 demonstrates that STAGE-BO achieves superior performance across additional metrics. While qPOTS remains a strong competitor due to its focus on maximizing diversity, it fails to achieve comprehensive coverage of the entire Pareto front; this limitation is evidenced by its significantly higher IGD+ values and lower hypervolume compared to our approach.

For the constrained MOO problems, in addition to IGD+ and fill distance, we report the feasibility ratio in Figure 10, defined as the proportion of observed points that satisfy all physical constraints throughout the optimization process. The consistent high feasible ratios achieved by STAGE-BO demonstrate that it can effectively estimate both the constraints and objectives to identify the feasible region and locate the constrained Pareto front.

Lastly, we show the IGD+ and fill distance results on preference-aware MOO tasks in Figure 11. STAGE-BO achieves superior performance across additional metrics on all datasets.

### D.2. Additional benchmark experiments

We present additional experimental results for the unconstrained MOO benchmarks ZDT2 ($d = 8, m = 2$) and the Coil Compression Spring design problem ($d = 3, m = 2$) in Figure 12.

Across both synthetic and engineering-design tasks, our method achieves superior performance in terms of HV, IGD, IGD+ and fill distance. These results further validate the effectiveness of our gap-filling strategy in maintaining high solution diversity and rapid convergence toward the true Pareto front.

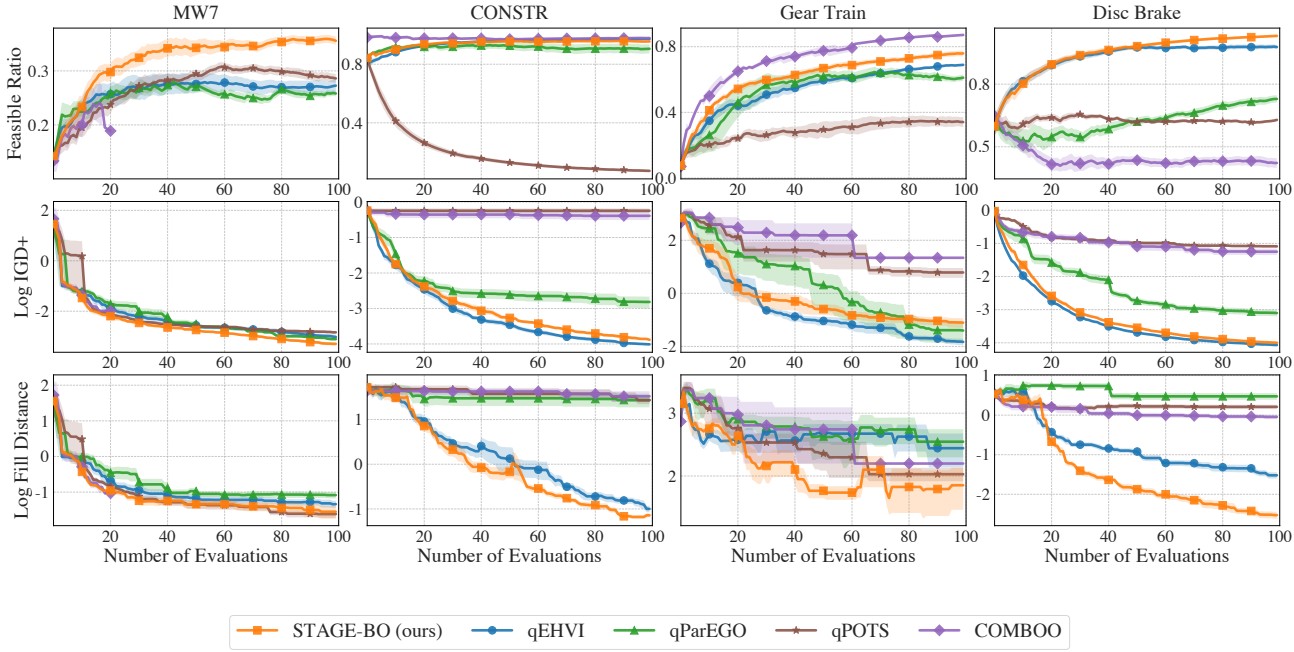

*Figure 10.* Comparison of our method with state-of-the-art baselines on one synthetic and three real-world benchmark **constrained MOO problems**. The first row reports feasible ratio, the second row reports IGD+, and the last row reports fill distance. Overall, our method achieves comparable or superior performance.

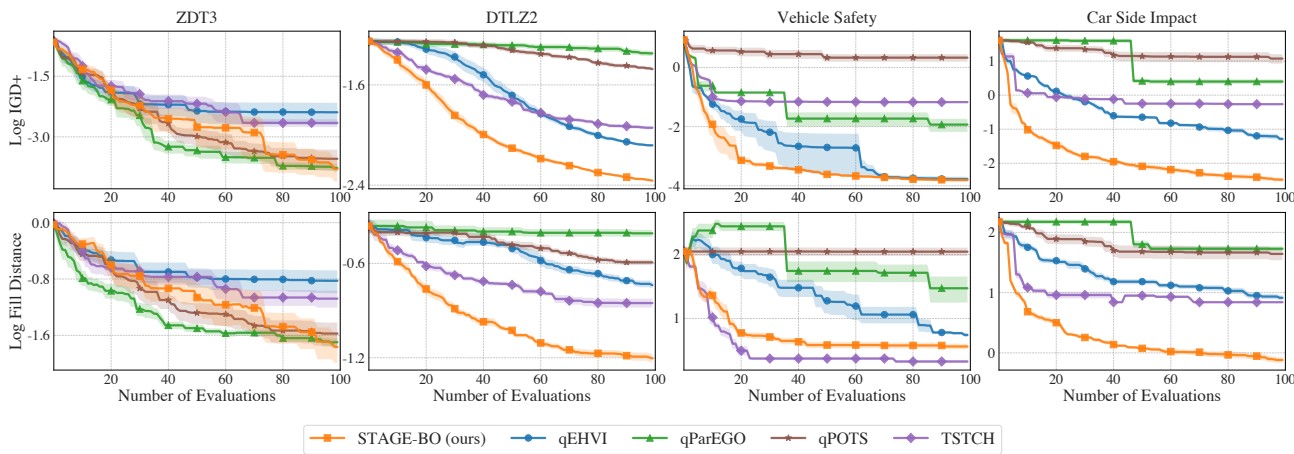

*Figure 11.* Comparison of our method with state-of-the-art baselines on two synthetic and two real-world benchmark **MOO problems with preferred regions**. The first row reports hypervolume, and the second row reports IGD. Overall, our method consistently outperforms baselinese in terms of IGD+ and fill distance.

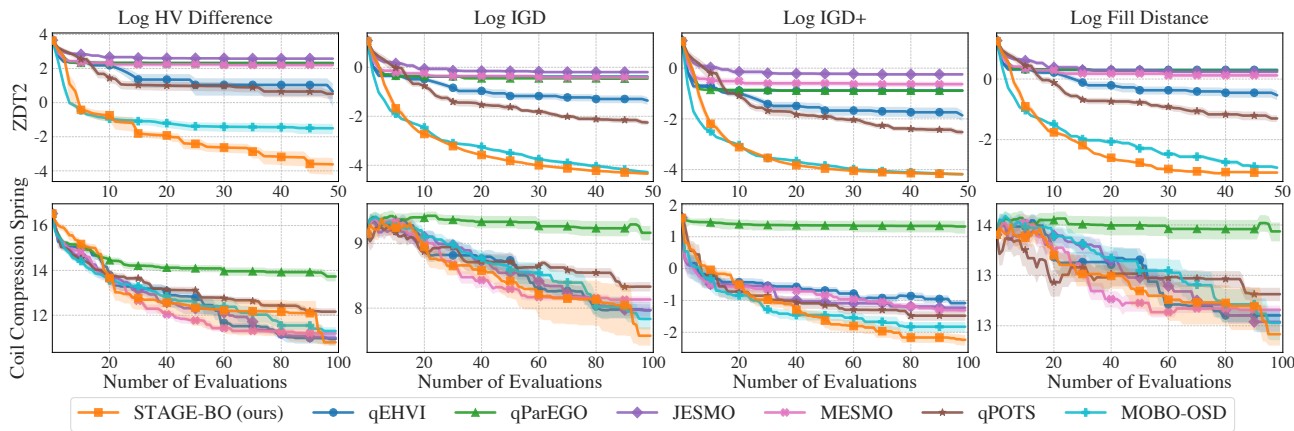

*Figure 12.* Comparison of our method with state-of-the-art baselines on one synthetic and one real-world benchmark **MOO problems**. The first row reports the results for ZDT2, and the second row reports results for Coil compression spring design. Overall, our method achieves comparable or superior hypervolume relative to the baselines and consistently outperforms them in terms of IGD and fill distance.

### D.3. Additional preferred-region experiments

We present additional experiments for the preference-based MOO benchmarks by varying the ROI—different bounds on ZDT3 ($d = 2, m = 2$). The performance results can be found in Figure 13, which show that STAGE-BO performs consistently better than all baselines. We additionally compare the empirical observed Pareto fronts of STAGE-BO against the other baselines in Figures 14 to 16, demonstrating it recovers the preferred Pareto front better than the baselines.

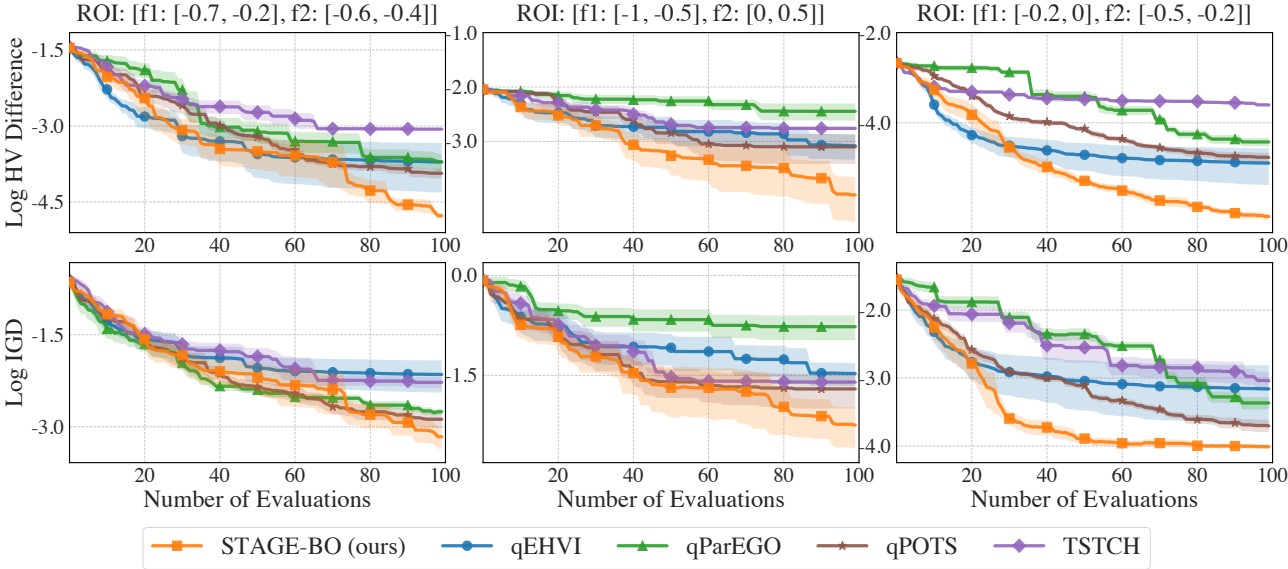

*Figure 13.* Comparison of STAGE-BO and baseline methods on three different preferred regions of the ZDT3 benchmark. The first row reports HV difference and the seoncd row reports IGD. Across all three regions, STAGE-BO consistently achieves the lowest HV difference and IGD, demonstrating the superior performance regardless of the location of regions.

## E. Computational Complexitye

We provide the theoretical computational complexity analysis as follows. Let $N_t$ denote the number of observations at step $t$, $d$ the input dimension, $m$ the number of objectives, $R$ the number of spectral features used in Thompson sampling, $P$ the NSGA-II population size, $G$ the NSGA-II generations. The per-iteration cost of STAGE-BO breaks down as follows: The GP training cost is well-known to scale cubically with the number of training samples, resulting in a computational complexity of $O(mN_t^3)$. We use Matheron construction for Thompson sampling, which results in the complexity of $O(m(dRN_t + N_t^2))$.

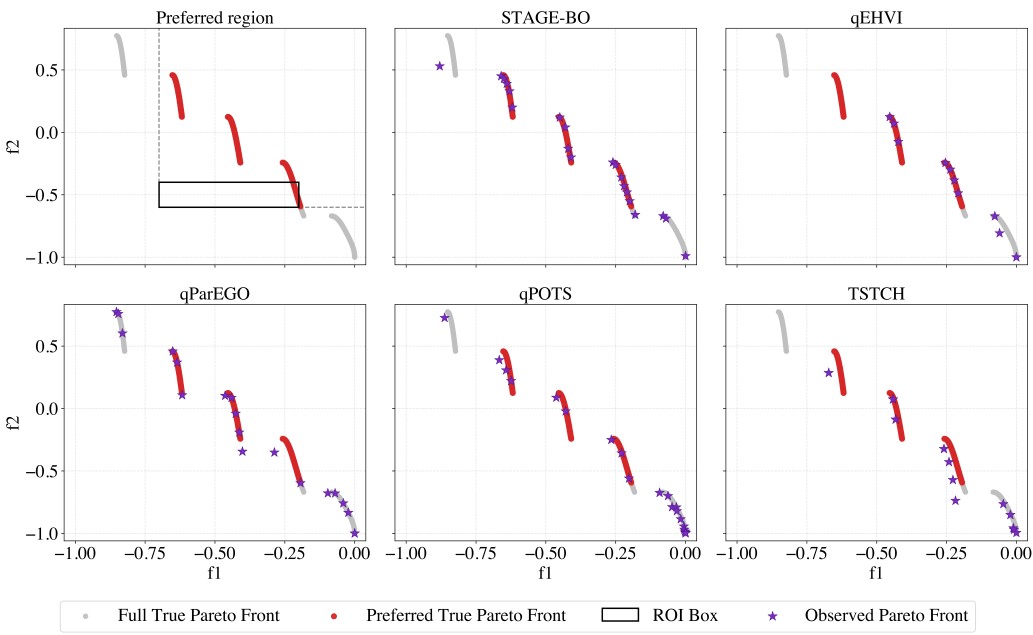

*Figure 14.* Comparison of observed Pareto fronts (purple stars) performed by different algorithms on the ZDT3 problem after 100 evaluations. The preferred region is set on for $f_1, [-0.7, -0.2]$ and for $f_2, [-0.6, -0.4]$. Grey points show the full Pareto front and red points present the preferred region. STAGE-BO can cover the preferred Pareto front more widely and uniformly.

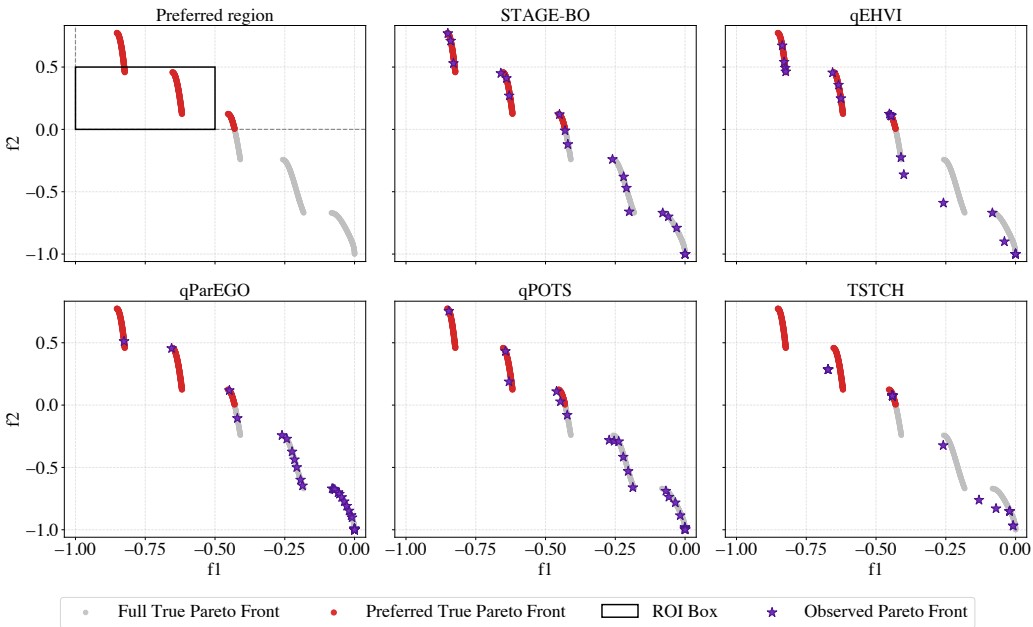

*Figure 15.* Comparison of observed Pareto fronts (purple stars) performed by different algorithms on the ZDT3 problem. The preferred region is set on for $f_1, [-1, -0.5]$ and for $f_2, [0, 0.5]$. Grey points show the full Pareto front and red points present the preferred region. STAGE-BO can cover the preferred Pareto front more widely and uniformly.

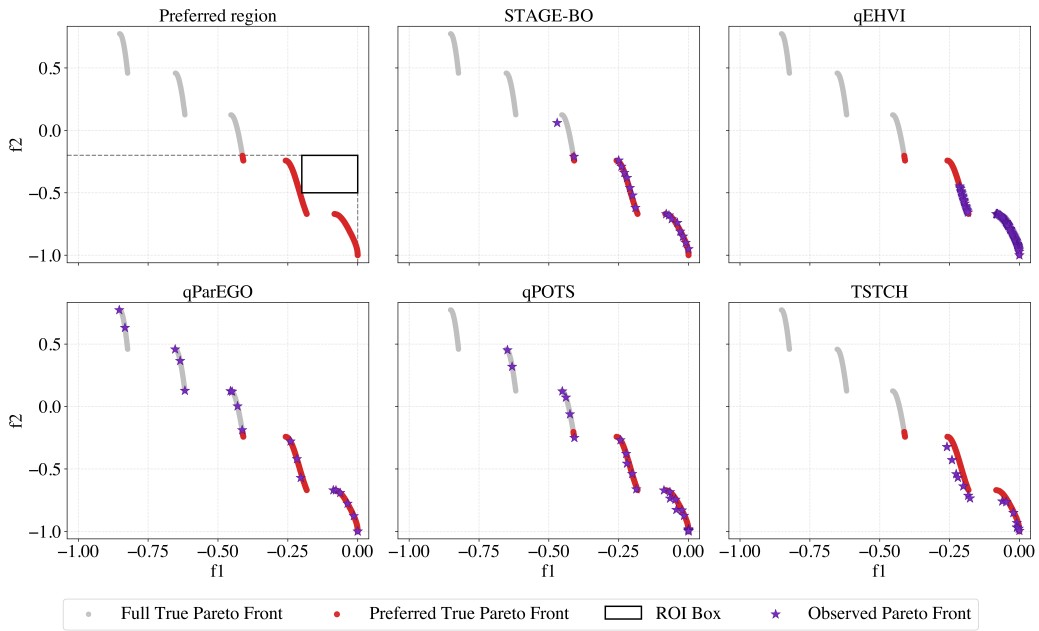

*Figure 16.* Comparison of observed Pareto fronts (purple stars) performed by different algorithms on the ZDT3 problem. The preferred region is set on for $f_1, [-0.2, 0]$ and for $f_2, [-0.5, -0.2]$. Grey points show the full Pareto front and red points present the preferred region. STAGE-BO can cover the preferred Pareto front more widely and uniformly.

The computational complexity for solving NSGA-II is $O(mGP^2 + mdGP(R + N_t))$, which comprises two dominated components: the non-dominated sort $O(mGP^2)$ and point evaluation $O(mdGP(R + N_t))$. The runtime complexity for fill distance between P predicted Pareto front and $N_t$ observations is $O(mPN_t)$ and for cEI optimization is $O(mN_t^2)$. Putting the pieces together, the overall dominated term is $O(mN_t^3)$ from GP fitting. When $N_t$ is small (e.g., $N_t < 200$), the NSGA-II term usually dominates. Importantly, all terms are polynomial in the number of objectives and decision variables. By contrast, the computational complexity for qEHVI is $O(mN_t^3 + mK)$, where $O(mN_t^3)$ is for the GP training and $O(mK)$ is for the box decomposition of the Pareto front. The number of hyperrectangles $K$ is super-polynomial in $m$. This is precisely why qEHVI becomes intractable for $m > 4$.

Additionally, we compare the empirical runtime of STAGE-BO against baselines across the unconstrained MOO tasks, with results summarized in Figure 17. STAGE-BO demonstrates high efficiency for problems with fewer than four objectives ($m < 4$). Furthermore, unlike many hypervolume-based methods that become computationally prohibitive as the objective space expands, our framework remains computationally tractable for higher-dimensional objective spaces ($m \geq 4$), maintaining a consistent per-iteration cost.

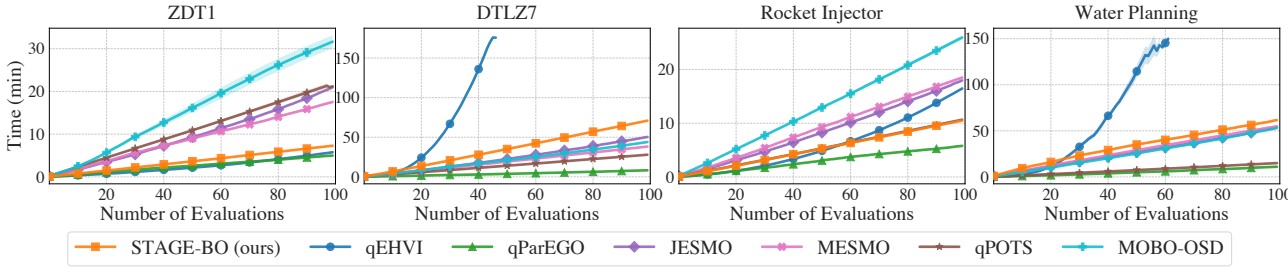

*Figure 17.* Computation time for STAGE-BO and all baselines. Our method shows high efficiency in low dimensions when $m \leq 4$ and remains computationally tractable for higher-dimensional objective spaces ($m \geq 4$).

# F. Ablation Study

We conduct ablation studies to validate the core components of STAGE-BO: effectiveness of Thompson sampling, ablation on clipping, effectiveness of targeted gap-filling, and robustness of objective selection.

## F.1. Effectiveness of Thompson Sampling.

As presented in Figure 18, replacing the Thompson sample in STAGE-BO with the posterior mean would have the negative consequence. This is expected: the posterior mean is overly greedy, suppressing the uncertainty-driven variability that makes Thompson sampling effective for exploration.

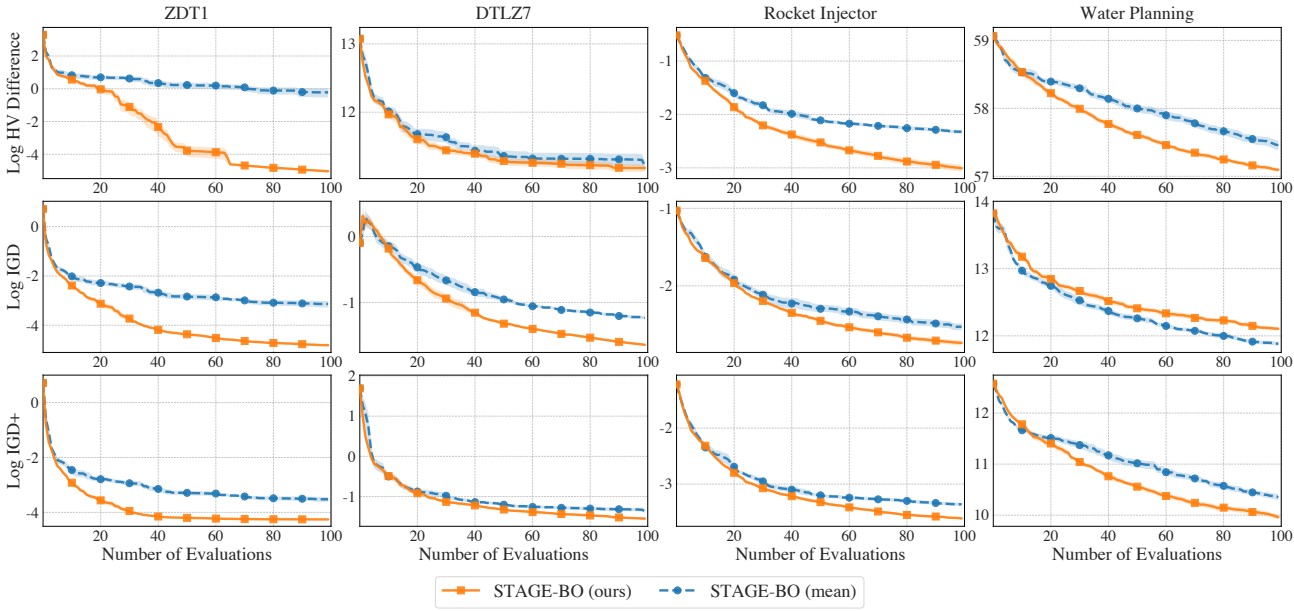

*Figure 18.* Ablation study on replacing the single Thompson sample in STAGE-BO with the posterior mean for unconstrained MOO. The three rows report log HV difference, log IGD and log IGD+, respectively. The Thompson-sampling variant performs better in most cases. On Water Planning, the posterior-mean variant is slightly better in IGD, while ours remains better in HV and IGD+. This suggests that Thompson sampling better exploits posterior uncertainty, whereas the posterior mean is more greedy and ignores uncertainty.

## F.2. Ablation of Clipping.

We conducted an ablation study comparing STAGE-BO with and without clipping. In practice, clipping arises primarily in early iterations when the posterior is uncertain and the sampled front overshoots the currently attainable region. As the number of observations grows and the posterior concentrates, the sampled front increasingly aligns with the true front and clipping is triggered less frequently. Results in Figures 19 to 21 show that on most benchmarks the two variants perform comparably, confirming that clipping acts primarily as a stabilizer. On a subset of benchmarks, clipping leads to measurable improvements, suggesting that the larger feasible region induced by clipping benefits optimization.

## F.3. Effectiveness of Targeted Gap-Filling.

The left panels of Figure 22 compare the STAGE-BO pipeline on ZDT1 against two variants:

1. **Direct Sampling (without cEI):** We bypass the acquisition optimization and directly query the maxmin target $\mathbf{Y}_c$. This evaluates whether cEI provides a necessary push toward the true front beyond the raw posterior samples.

2. **Random Constraints (No Maxmin):** We replace the geometric target $\mathbf{Y}_c$ with a randomly selected coordinate within the observed front to verify the necessity of explicit gap-filling via fill-distance minimization. To ensure this randomly selected target defines a non-empty feasible region, we employ a Lexicographical Constraint-Setting procedure: we

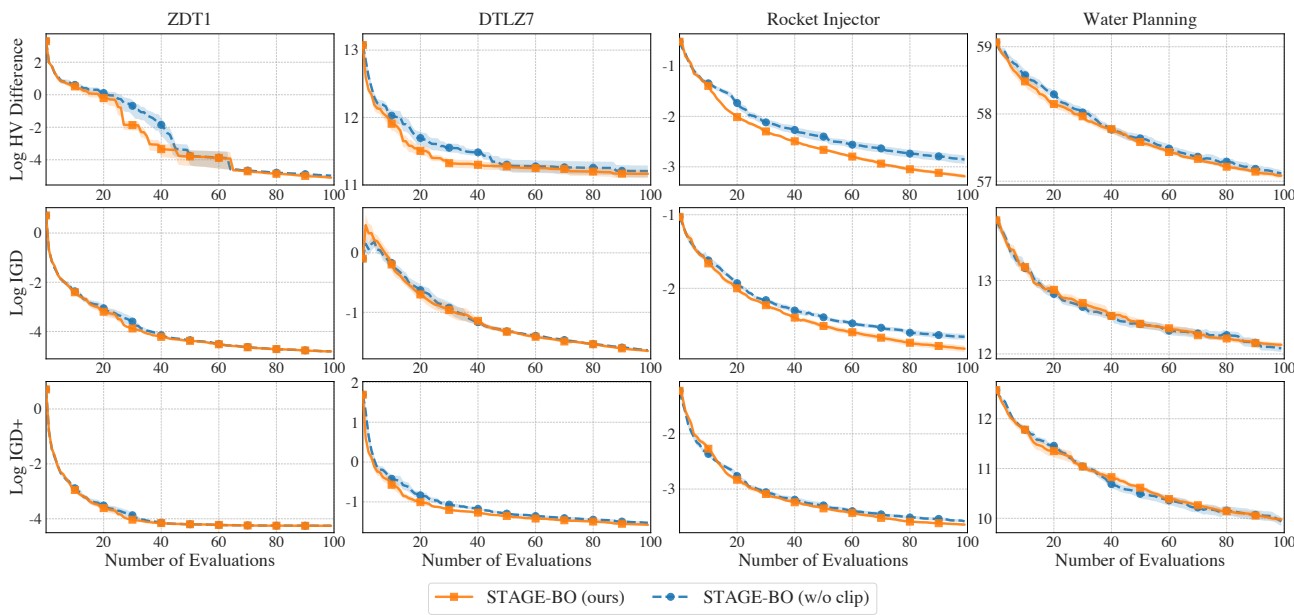

*Figure 19.* Ablation study on clipping in STAGE-BO for unconstrained MOO. The three rows report log HV difference, log IGD and log IGD+, respectively. Clipping is designed as a numerical stabilizer and has little effect on most datasets, but consistently improves HV and IGD on Rocket Injector.

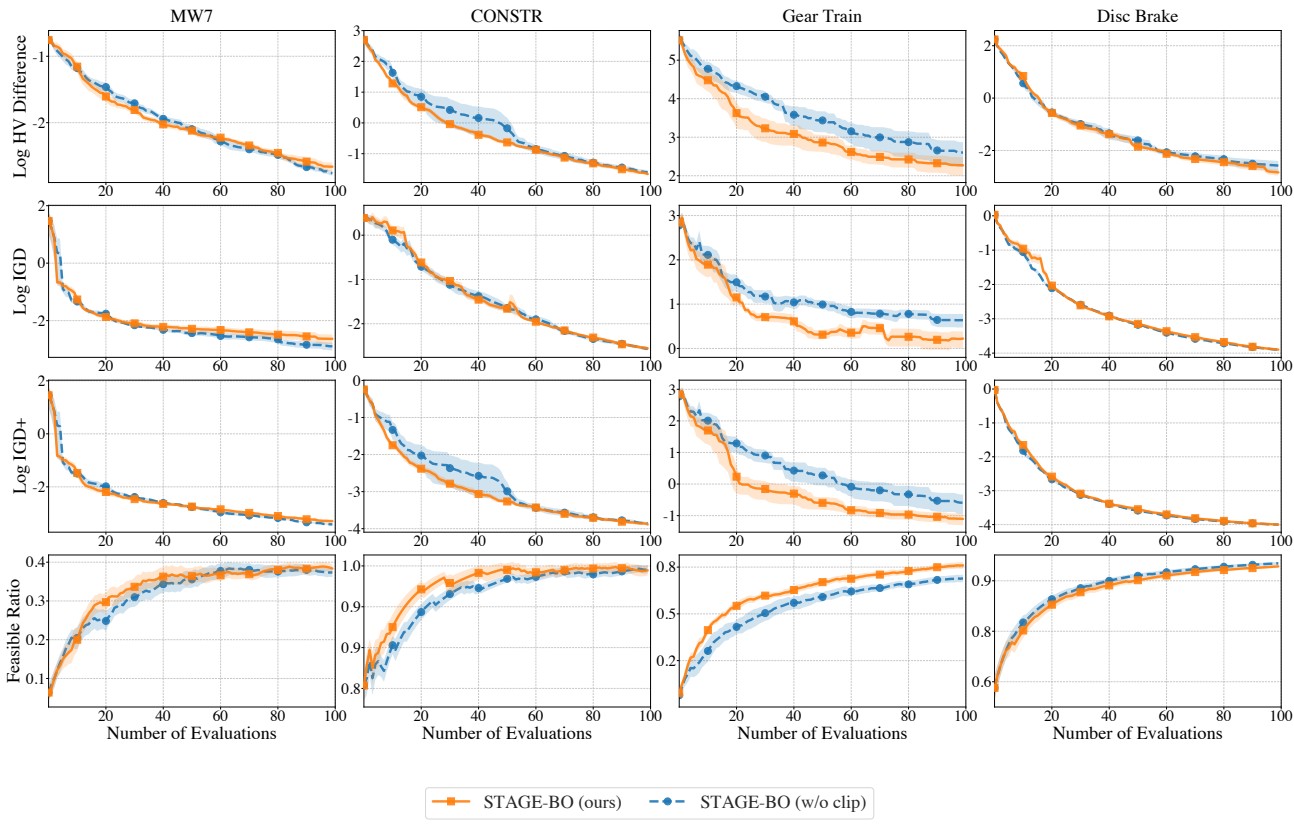

*Figure 20.* Ablation study on clipping in STAGE-BO for constrained MOO. The four rows report log HV difference, log IGD, log IGD+ and the feasible ratio of evaluated solutions under constraints, respectively. Clipping is designed as a numerical stabilizer and has little effect on most datasets, but leads to clear gains on Gear Train.

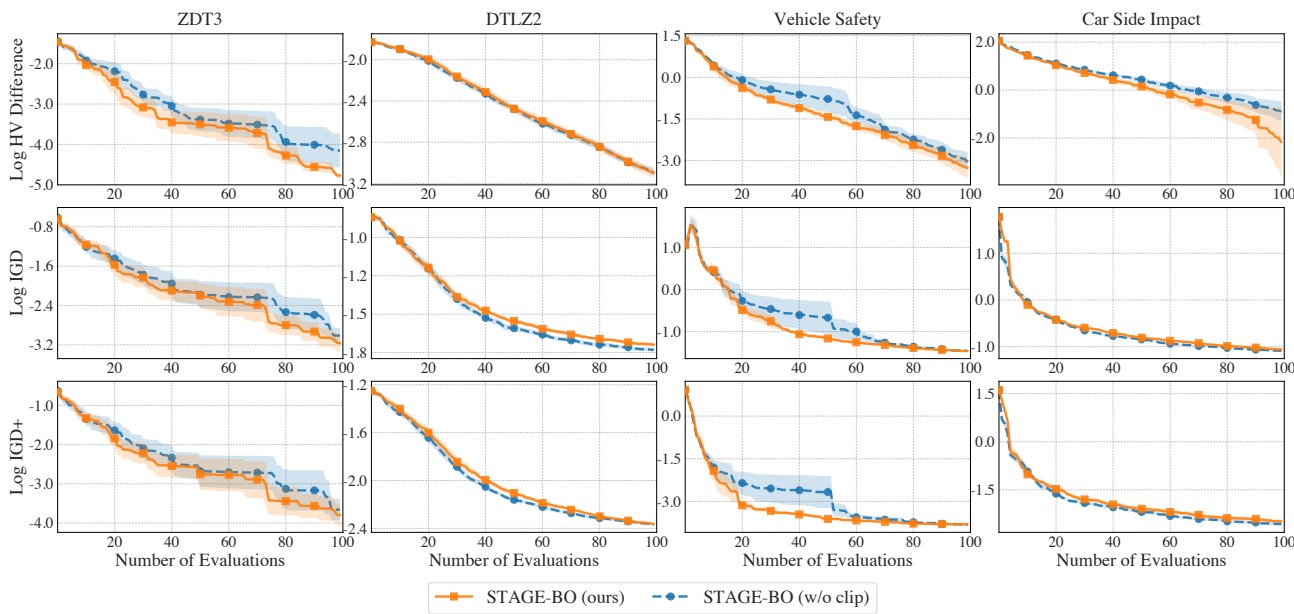

*Figure 21.* Ablation study on clipping in STAGE-BO for preference-based MOO. The three rows report HV difference, IGD and IGD+, respectively. The clipped and unclipped variants perform similarly across all datasets, indicating that clipping acts as a numerical stabilizer.

sequentially determine the thresholds $\varepsilon_j$ by validating feasibility for each objective in order $(f_1, f_2, \ldots, f_m)$, ensuring that each subsequent constraint remains reachable given the previous ones.

The results support that the combination of maxmin targeting and cEI optimization is essential for achieving superior convergence and uniform coverage.

### F.4. Robustness of Objective Selection.

The right panels of Figure 22 investigate the schedule for selecting the primary objective $f_k$. We compare our default round-robin schedule against two alternatives:

1. **Random Optimization:** The objective $f_k$ is chosen at random in each iteration.

2. **Feasible Optimization:** To maximize the potential feasible region for the acquisition solver, we identify the objective $f_k$ with the minimum coordinate distance between $\mathbf{Y}_c$ and its nearest observed neighbor, thereby optimizing the dimension with the most "crowded" constraints.

The results demonstrate that STAGE-BO is insensitive to the objective selection strategy.

## G. Baselines Implementation

We implemented STAGE-BO and all baselines in Python (version 3.10). The detailed implementation are as follows.

**STAGE-BO** For the surrogate model, we implement the GPs via GPyTorch (Gardner et al., 2018) and BoTorch (Balandat et al., 2020). We employ a Matern 5/2 kernel with ARD length-scales. The Gaussian likelihood is modeled with homoskedastic noise, and model hyperparameters are optimized by maximizing the Sum Marginal Log-Likelihood. During fitting, a Cholesky jitter of $10^{-3}$ is applied to maintain numerical stability. For the Thompson sampling, we draw a sample path from each GP posterior using Matheron's rule (Wilson et al., 2020), which decomposes the posterior sample into a prior spectral sample and a data-dependent update term, yielding a continuous and differentiable approximation of each objective.

We set 300 population size and 50 max generations for the NSGA-II and use BoTorch to optimize cEI with 20 starters.

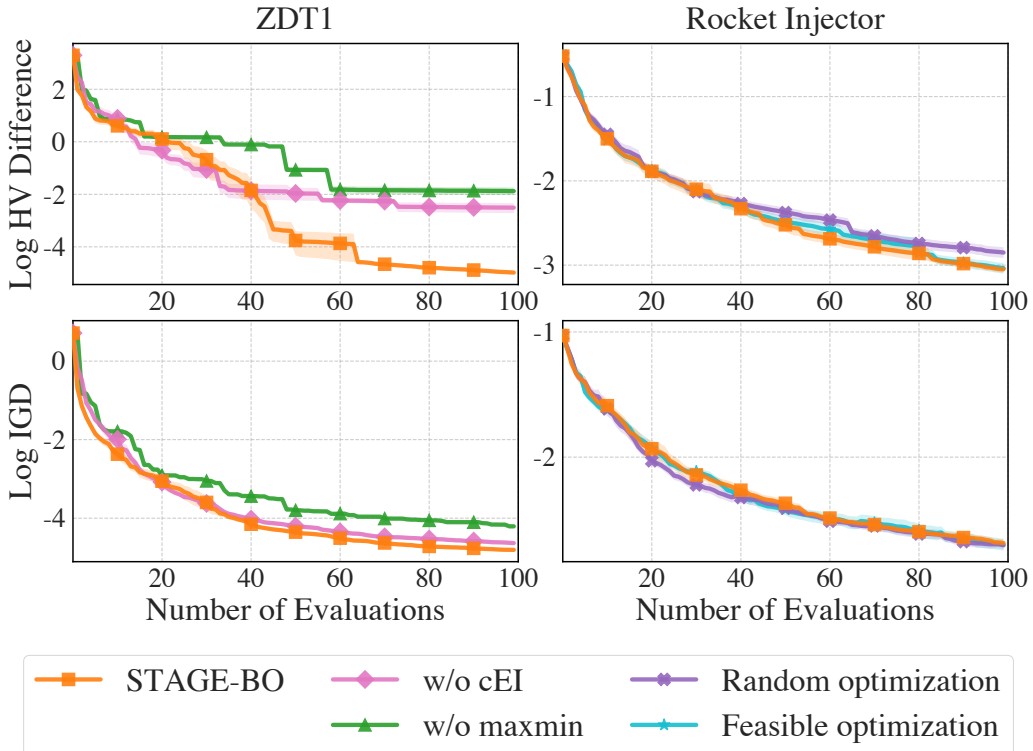

*Figure 22.* Ablation study on different parts of our method. The left panels show that cEI and computing maxmin to set the constraints are necessary. The right panels show that our method is robust to the strategy of picking the objective to optimize.

**qEHVI (Daulton et al., 2020)**   We use the default hyperparameter settings from the paper and the open-sourced implementation can be found at `https://github.com/pytorch/botorch`. In the preference-aware setting, qEHVI needs to be adapted. Simply setting the reference point at the corner of the ROI hypercube is not always appropriate. As illustrated in Figure 3, the user-defined ROI may be overly ambitious and lie entirely beyond the attainable front. In this case, setting the reference point at the corner of the ROI hypercube means no observed solution dominates the ROI corner, and qEHVI accumulates zero hypervolume improvement throughout the entire optimization and becomes ineffective. To avoid this, we adaptively set the reference point for qEHVI in a way consistent with our framework. We construct the qEHVI reference point from a Thompson-sampled path: we sample a posterior path over the design space, identify sampled points that fall inside the feasible preference region, and set the reference point to their coordinate-wise lower bounds. If no sampled point falls inside the preferred region, we fall back to the coordinate-wise lower bounds of the full sampled path.

**qParEGO (Knowles, 2006)**   qParEGO is a novel extension from ParEGO (Knowles, 2006) that is developed by Daulton et al. (2020) to leverage batch setting. We use the settings as follows: augmented Tchebychev scalarization (Nakayama et al., 2009) and EI acquisition function with gradient solver. We use the open-sourced implementation at `https://botorch.org/docs/tutorials`.

**MESMO (Wang & Jegelka, 2017)**   We use the default hyperparameter settings and the open-sourced implementation can be found at `https://github.com/pytorch/botorch`.

**JESMO (Tu et al., 2022)**   We use the default hyperparameter settings and the open-sourced implementation can be found at `https://github.com/pytorch/botorch`.

**qPOTS (Renganathan & Carlson, 2025)**   We use the default hyperparameter settings. This includes the NSGA-II hyperparameter settings. We use the open-sourced implementation at `https://github.com/csdlpsu/qpots`.

**MOBO-OSD (Ngo et al., 2025)** We use the default hyperparameter settings from the paper. This includes the number of points on approximated CHIM and the number of starting points when solving MOBO-OSD subproblem. We use the open-sourced implementation at `https://github.com/LamNgo1/mobo-osd`.

**COMBOO (Li et al., 2025)** We use the default hyperparameter setting from the paper. The open-sourced implementation can be found at `https://github.com/dancewithDianTong/COMBOO`.

**TSTCH (Paria et al., 2020)** We use the settings as follows: augmented Tchebychev scalarization (Nakayama et al., 2009) and Thompson sampling acquisition function. We implement this with BoTorch (Balandat et al., 2020).

**MOEA/D-EGO (Zhang et al., 2009)** We use the default hyperparameter setting from the paper. The open-sourced implementation can be found at `https://github.com/mobo-d/MOEAD-EGOO`.

**DirHV-EGO (Zhao & Zhang, 2023)** We use the default hyperparameter setting from the paper. The open-sourced implementation can be found at `https://github.com/mobo-d/DirHV-EGO`.

## H. Benchmark Problems

Here we present the benchmark problems used in unconstrained MOO, constrained MOO and preference-aware MOO.

*Table 2.* Benchmark problem settings and reference points in **unconstrained multi-objective problems**.

| Problem | d | m | Reference Point |
|---|---|---|---|
| ZDT1 | 10 | 2 | $(-11.0, -11.0)$ |
| ZDT2 | 8 | 2 | $(-11.0, -11.0)$ |
| DTLZ7 | 6 | 5 | $(-1.1, -1.1, -1.1, -1.1, -1.1)$ |
| Coil Compression Spring | 3 | 2 | $(-133.65, -9056129.08)$ |
| Rocket Injector | 4 | 3 | $(-0.96, -1.11, -1.08)$ |
| Water Planning | 3 | 6 | $(-84348.75, -1460.57, -3101483.5, -12442799.73, -67029.71, -1.59)$ |

*Table 3.* Benchmark problem settings and reference points in **constrained multi-objective problems**.

| Problem | d | m | C | Reference Point |
|---|---|---|---|---|
| MW7 | 4 | 2 | 2 | $(-1.2, -1.2)$ |
| CONSTR | 2 | 2 | 2 | $(-10.0, -10.0)$ |
| Disc brake design | 4 | 2 | 4 | $(-7.58, -7.0)$ |
| Gear train design | 4 | 2 | 1 | $(-7.4, -64.1)$ |

*Table 4.* Benchmark problem settings and reference points in **preference-aware multi-objective problems**.

| Problem | d | m | Preferred Region |
|---|---|---|---|
| ZDT3 | 2 | 2 | $(-1, -1)$ |
| DTLZ2 | 6 | 5 | $(-0.8442, -0.8999, -0.8358, -0.8710, -0.8553)$ |
| VehicleSafety | 5 | 3 | $(-1680, -7.0, -0.26)$ |
| CarSideImpact | 7 | 4 | $(-23.01, -4.43, -13.09, -9.47)$ |

