# OpenReview forum: "Multi-Objective Bayesian Optimization via Adaptive $\varepsilon$-Constraint Decomposition"
_ICML.cc/2026/Conference — ICML 2026 regular_

### Official Review · Reviewer_J1pk · 2026-03-05

**Soundness:** 3
**Presentation:** 3
**Significance:** 1
**Originality:** 1
**Overall Recommendation:** 3
**Confidence:** 4

**Summary:**

The authors focus on multi-objective bayesian optimization, specifically in enhancing the coverage of solutions on the Pareto frontier. The authors do so by explicitly targeting the under-explored regions of the Pareto front, which they can then transform into a sequence of inequality-constrained subproblems. The authors also apply the method to constrained and preference-based multi-objective optimization settings and evaluate on several synthetic and real-world problems.

**Compliance With Llm Reviewing Policy:**

Affirmed.

**Final Justification:**

I thank the authors for their responses, and I appreciate the additional commentary. I however still remain on the skeptical side about the significance of the work, regarding the need for obtaining uniform coverage of the entire Pareto frontier. In practice, it seems a decision maker would often have some sort of prior over certain regions of the Pareto frontier. The authors provide some examples pertaining to neural architecture search and molecular discovery in their response. In such situations, especially neural architecture search, obtaining just a single point on the Pareto frontier is typically very expensive (computationally and maybe financially). Ensuring for more uniform coverage over the entire frontier, especially within regions showing marginal gains, seems like a rather impractical approach. In general, my prior assessment is generally reinforced and, as such, I maintain my score.

**Key Questions For Authors:**

1. Why are evolutionary algorithms such as NSGA-II used for generating a discrete approximation of the front? Is this general beyond more than just evolutionary algorithms?
2. Is STAGE-BO specific to the proposed acquisition function? (EI + PoF)
3. Variations on the preferred region for those experiments could be helpful, e.g. specifying different preferred regions
4. The IGD metric seems to assume you have access to the true Pareto front. How would this work in a true real-world scenario? Since I'm assuming in the real-world settings the paper evaluates on, the true Pareto front is provided or very easily accessible.

**Limitations:**

Yes.

**Strengths And Weaknesses:**

Strengths:
- Paper is well-organized and structured
- Presented results from a diverse set of experimental settings often surpass those of the baselines
- Different settings are discussed/presented, e.g. preference-aware, constrained

Weaknesses:
- Perhaps the largest weakness is that it is unclear whether the paper addresses an important or relevant problem. I don't think the manuscript does particularly well at exhibiting this. For instance, the Introduction mentions that "HV maximization heavily concentrates solutions in steep knee regions". Why is that an issue? Isn't the steep knee region where most of the critical tradeoffs occur and, as a result, should be more heavily sampled? The other regions with flat tradeoffs are not as interesting and can be easily discerned with a smaller set of samples, as I understand it. Please better motivate this either with a real-world application in the Introduction (beyond what is already presented) or via a user study, etc.
- Some of the writing is a bit unclear. For instance, IGD should be defined in Figure 1. In the \epsilon-constraint Decomposition section, in 4.1, it's unclear how Y_c is used in determining the constraints. This isn't mentioned until the section afterwards, Acquisition Optimization. Also, why is the full algorithm only presented in the Appendix Section A? I think it should be in the main text for clarity.
- Some design decisions can be better motivated (see below for questions)
- Some missing related work that I think should be discussed:
Edward Chen, Natalie Dullerud, Thomas Niedermayr, Elizabeth Kidd, Ransalu Senanayake, Pang Wei Koh, Sanmi Koyejo, Carlos Guestrin. "MoSH: Modeling Multi-Objective Tradeoffs with Soft and Hard Bounds", 2024. https://arxiv.org/abs/2412.06154

---

> ### Author Rebuttal · Authors · 2026-03-31
>
> Thank you for the thoughtful feedback and for highlighting the broad experimental scope. We address the concerns below.
>
> **Weaknesses:**
>
> **Why coverage matters / significance of the problem.**
> On Pareto optimality: By definition, every point on the Pareto front is mathematically equivalent — there is no ordering among Pareto-optimal solutions without additional preference information. HV maximization implicitly imposes a preference by assigning higher acquisition value to solutions near steep knee regions, which contradicts the foundational assumption of Pareto optimality. Whether knee regions are "more important" is precisely the kind of preference judgment that should belong to the decision-maker, not the optimization algorithm.
>
> On practical motivation: We do not argue that knee regions are unimportant — they are often highly valuable. The issue is that HV-based methods concentrate on knee regions by default, even when the decision-maker's preferences may lie elsewhere. In many real MOBO applications, the purpose is to present a diverse set of Pareto-optimal tradeoffs to downstream stakeholders who may later impose preferences, practical constraints, or domain considerations not captured in the model. Systematically under-covering large portions of the front can hide solutions that would have been selected under those downstream preferences. Uniform coverage is therefore the appropriate objective whenever the goal is broad tradeoff discovery rather than targeted single-region exploitation — which is arguably the more common MOBO use case. We will clarify these in the Intro.
>
> **Clarity.** Thank you for the specific suggestions.We will explicitly introduce IGD before using it in Fig 1 and will better rephrase our method, especially the target identification steps. We placed the full version of our algorithm in the appendix for space reasons, but we will move a concise version into the main text for clarity.
>
> **Related work (MoSH).** Thank you for pointing us to MoSH. MoSH's motivation strongly supports the importance of our work: rather than asking decision-makers to inspect the entire Pareto front, it allows them to specify acceptable regions via soft and hard bounds, then returns a concentrated set of Pareto-optimal solutions within that preferred region. This is precisely the preference-aware setting addressed in Section 4.3, where STAGE-BO accepts ROI bounds directly as constraints on the inner MOO problem. We will discuss this connection explicitly in Related Work.
>
> **Questions:**
>
> **1. Why NSGA-II.** NSGA-II is used as a practical implementation choice for the inner MOO problem, not as a fundamental component of the framework. The only requirement on this inner solver is that it can produce a reasonable discrete approximation of the sampled Pareto front efficiently. Any multi-objective solver satisfying this requirement is a valid substitute. For instance, random scalarization-based methods, or first order approximation used in MOBO-OSD could all serve this role. We chose NSGA-II because it is widely used [1-2], well-tested, and straightforwardly applicable across varying numbers of objectives.
>
> **2. Is STAGE-BO specific to EI + PoF?** No. In the paper we use constrained EI because it is a natural and simple solver for that subproblem, but the framework is modular and other constrained BO acquisition functions, such as constrained UCB, or more advanced optimizers, could also be used. We discuss this design choice in Appendix C, including why cEI is a good fit here and why some alternatives, such as trust-region-based constrained optimization methods, may be less suitable in our setting. Our main contribution is not the particular constrained acquisition function, but the way we transform the MOO problem into a sequence of adaptive single-objective constrained subproblems through posterior-guided target identification.
>
> **3. Preferred-region variations.** We ran additional preferred-region experiments to demonstrate that STAGE-BO is agnostic to the location of preferred regions: https://anonymous.4open.science/r/ICML-1EEC/Different_preferred_regions.md  STAGE-BO performs consistently better than all baselines.
>
> **4. IGD computation.** We would like to highlight that our method does not require access to the true Pareto front; we use IGD only as an evaluation metric for benchmark coverage. We additionally present a real-world application: please refer to the responses to Reviewer **yvUL** for details. In this case, since the true Pareto front is unknown, IGD is not available and our method still has superior performance in HV.
>
> [1] Bradford, E., et al. (2018). Efficient multiobjective optimization employing Gaussian processes, spectral sampling and a genetic algorithm.
>
> [2] Renganathan, S. et al.. (2025). qpots: Efficient batch multiobjective bayesian optimization via pareto optimal thompson sampling.

---

> > ### Author Rebuttal · Reviewer_J1pk · 2026-04-04
> >
> > Thank you for the response. I am still not convinced of the significance of this work (the W1 I pointed out initially). In your rebuttal, you mention "In many real MOBO applications, the purpose is to present a diverse set of Pareto-optimal tradeoffs ..." however I think the work could benefit greatly if you motivate it from the perspective of one (or more) specific/concrete real-world applications. More specifically, are there real-world problems where having access to a uniform coverage set actually helps with decision making? As I had mentioned previously, one would think that the regions outside of the knee points are not changing in metric values as much and, therefore, ensuring such regions are uniformly covered may actually be overwhelming the decision maker with too many points, or using unnecessary extra computation.
> >
> > I also agree with reviewer yvUL's point about having more real-world problems. The problems in RE are on the easier side and do not seem to illustrate any significance of this work. The privacy-preserving ML application is interesting, but, again, I think the importance here is convincing the reader why this approach to sampling beyond the knee points is important.

---

> > > ### Author Response · Authors · 2026-04-06
> > >
> > > We thank the reviewer for the continued discussion and address the concerns further below.
> > >
> > > **1. Why sampling beyond the knee points is important.**
> > >
> > > A well-covered Pareto front is valuable in many real applications, such as when the resulting Pareto front is later used by multiple downstream users with heterogeneous preferences, when post-hoc constraints may invalidate knee-region solutions, or when the front serves as a decision support artifact rather than a final answer.
> > >
> > > For example, in neural architecture search [1], after the Pareto front between accuracy and latency is computed, different teams select architectures under different deployment constraints. A knee solution may offer the best generic compromise, but an edge deployment with battery constraints may prioritize inference speed over accuracy, while a safety-critical deployment may favor a slower model to capture even a marginal gain in accuracy. In both cases, the preferred solution can lie well outside the knee region. A knee-concentrated front would fail to provide optimal options for one or both of these downstream uses.
> > >
> > > Similarly, in de novo polymer design [2] and molecular design [3], the goal is also to efficiently recover a diverse and dense set of Pareto-optimal candidates. It is especially highlighted in [3] that *"Hypervolume improvement can indicate the shift in the Pareto front, but other qualities can be of equal importance, including **the density of the Pareto front**. In molecular discovery, it is beneficial to discover a dense Pareto front. Naturally, some molecules that are predicted to perform well will not validate experimentally, and having a denser population to sample from will increase the probability of finding true hits."*
> > >
> > > These examples support the motivation of STAGE-BO. In many applications, additional constraints may appear later, downstream users may have different priorities between objectives, and even a very small improvement in a primary objective may justify a large sacrifice in another. As a result, regions beyond the knee can be just as decision-relevant as the knee itself.  When preferences are unknown at optimization time, the algorithm should not internally discard regions of the front based on an implicit assumption about what decision-makers will care about. A well-covered Pareto front preserves these alternatives for downstream decision-making.
> > >
> > >
> > > **2. On the implicit bias of HV-based methods.**
> > >
> > > As proved in [4], HV maximization imposes an implicit preference not only through front geometry (concentrating near steep regions) but also through reference point placement, which can further bias the preferred region toward one objective over another.
> > >
> > > If the decision makers have already specified the preference towards the knee region, then the optimization should focus there. The key point is that, in many MOO applications, such preferences are not known at optimization time, or differ across downstream users. In those settings, the goal of MOO is often to generate a representative set of Pareto-optimal tradeoffs that can support later decision-making, rather than to commit in advance to one region of the front.
> > >
> > >
> > > **3. On diversity significance.**
> > >
> > > As we discussed in the Introduction and Related Work in our paper, recent diversity-guided MOBO methods [5-8] reflect growing interest in recovering not only high-quality Pareto solutions, but also well-covered trade-off sets. We believe STAGE-BO makes a principled contribution to this direction: **it achieves strong performance in terms of both IGD and HV, while avoiding the need for HV computation.**
> > >
> > > **4. Real-world application.**
> > >
> > > We would like to stress that our evaluation is on par with the norm in the literature and is at least as comprehensive as [5-8], even before we added the results on the privacy-preserving application, and STAGE-BO consistently performs strongly across the board.
> > >
> > >
> > >
> > > [1] Mao, S., et al. (2024). RAM-NAS: Resource-aware Multiobjective Neural Architecture Search Method for Robot Vision Tasks. IROS.
> > >
> > > [2] Jablonka, K. M., et al. (2021). Bias free multiobjective active learning for materials design and discovery. Nature communications.
> > >
> > > [3] Fromer, J. C., et al. (2023). Computer-aided multi-objective optimization in small molecule discovery. Patterns.
> > >
> > > [4]Auger, A., et al. (2009) Theory of the hypervolume indicator: optimal μ-distributions and the choice of the reference point.
> > >
> > > [5] Konakovic Lukovic, M., et al. (2020). Diversity-guided multi-objective Bayesian optimization with batch evaluations. NeurIPS.
> > >
> > > [6] Ahmadianshalchi, A., et al. (2024). Pareto front-diverse batch multi-objective Bayesian optimization. AAAI.
> > >
> > > [7] Renganathan, S. A., et al. (2025). qpots: Efficient batch multiobjective Bayesian optimization via pareto optimal thompson sampling. AISTATS.
> > >
> > > [8] Ngo, L., et al. (2025). MOBO-OSD: Batch Multi-Objective Bayesian Optimization via Orthogonal Search Directions. NeurIPS.

---

### Official Review · Reviewer_XQoK · 2026-03-05

**Soundness:** 3
**Presentation:** 3
**Significance:** 2
**Originality:** 2
**Overall Recommendation:** 3
**Confidence:** 3

**Summary:**

The paper introduces STAGE-BO, a method designed to address the limitations of standard multi-objective Bayesian optimization (MOBO). Traditional methods often rely on hypervolume (HV) maximization, which is computationally expensive for many objectives and biased toward steep "knee" regions of the Pareto front. STAGE-BO instead identifies the largest geometric "gaps" in the current Pareto front approximation and targets these voids using the $\epsilon$-constraint method. By transforming the multi-objective problem into a sequence of constrained single-objective subproblems solved via constrained expected improvement, the method achieves more uniform Pareto coverage (lower IGD) while remaining computationally tractable as the number of objectives increases.

**Compliance With Llm Reviewing Policy:**

Affirmed.

**Final Justification:**

I keep my score as the significance of this work is not sufficient for presentation at this conference.

**Key Questions For Authors:**

1. How does the number of spectral samples used to approximate the GP posterior (Equation 6) affect the quality of the "cheap" Pareto front and the subsequent target identification?
2. In Equation 10, you employ a clipping strategy to avoid empty feasible regions. Does this strategy ever lead to "over-exploration" in regions where the objectives are fundamentally limited, and how does it affect convergence speed in such cases?
3. While your method avoids HV costs, it requires solving an internal MOO problem with NSGA-II at every iteration. How does the overhead of this internal step scale relative to the overall BO iteration as the number of objectives and decision variables increases?

**Limitations:**

Yes

**Strengths And Weaknesses:**

#### Strengths

- The methodology is built on established theoretical foundations, such as the $\epsilon$-constraint method and fill distance, to ensure that targeting gaps leads to uniform coverage.
- The empirical evaluation is extensive, utilizing six synthetic benchmarks and multiple real-world problems (e.g., rocket injector, water planning) across unconstrained, constrained, and preference-aware settings.
- The paper is well-structured, clearly motivating the bias of HV-based methods with visual examples (Figure 1) and providing a detailed algorithmic framework in Section 3 and Appendix A.



#### Weaknesses

- The gap-filling mechanism relies on Euclidean distances in the objective space, making it highly sensitive to the relative scales of different objectives. Without a robust dynamic normalization strategy, the algorithm will be biased toward filling gaps in objectives with larger magnitudes, leading to an unbalanced Pareto front.
- The evaluation primarily compares against hypervolume-based methods but neglects decomposition-based approaches (e.g., MOEA/D-EGO or ParEGO). Since decomposition methods are the standard for high-dimensional objective scaling, their absence makes the claimed advantages of STAGE-BO less convincing.
- While avoiding hypervolume calculations, the method requires running an internal multi-objective solver (NSGA-II) in every iteration to identify target gaps. The paper lacks a detailed wall-clock time analysis, leaving concerns about the total computational cost and hyperparameter sensitivity of this nested optimization step.

---

> ### Author Rebuttal · Authors · 2026-03-31
>
> Thank you for the detailed and constructive feedback. We address the concerns below.
>
> **Weaknesses:**
>
> **Normalization.** Thanks for raising this important point. We want to clarify that target identification and fill distance computation are performed in a dynamically normalized objective space throughout our implementation. At each iteration, normalization bounds are derived from the current observations and applied to both the sampled Pareto front and the current observations before distance computation. This ensures that gap detection is not biased toward objectives with larger magnitudes. The superior performance of STAGE-BO on real-world benchmarks with heterogeneous objective scales provides empirical evidence that our method handles scale differences robustly. We will clarify these implementation details in revision and release the code.
>
> We also want to note that the performance metrics based on the distance (IGD, IGD+) are reported in the raw objective scale in our paper, consistently with standard practice in the MOBO literature. We have additionally verified that conclusions remain unchanged when performance metrics are computed in the normalized objective space — our method consistently outperforms baselines in both cases: https://anonymous.4open.science/r/ICML-1EEC/Normalized_results.md.
>
> **Baselines.** We already compared ParEGO in all the experiments and additionally added the two decomposition-based baselines. Please see the response to reviewer **yvUL** for detailed discussion.
>
> **Runtime complexity.** Figure 8 in our paper presents wall-clock time comparisons. STAGE-BO remains computationally efficient for the number of objectives m<=4 objectives and, crucially, maintains a tractable per-iteration cost as m grows — unlike HV-based methods whose cost becomes prohibitive for m>4.
>
> Here we present the theoretical analysis: Let N_t denote the number of observations at step t, d the input dimension, m the number of objectives, R the number of spectral features used in Thompson sampling. P the NSGA-II population size, G the NSGA-II generations. The per-iteration cost of STAGE-BO breaks down as follows:
>
> The GP training cost is well-known to scale cubically with the number of training samples, resulting in a computational complexity of O(mN_t^3).
>
> We use Matheron construction for Thompson sampling, which results in the complexity of O(m(dRN_t+N_t^2)).
>
> The computational complexity for solving NSGA-II is O(mGP^2+mdGP(R+N_t)), which comprises two dominated components: the non-dominated sort O(mGP^2) and point evaluation O(mdGP(R+N_t)).
>
> The runtime complexity for fill distance between P predicted Pareto front and N_t observations is O(mPN_t) and for cEI optimization is O(mN_t^2).
>
> Putting the pieces together, the overall dominated term is O(mN_t^3) from GP fitting. When N_t is small (e.g., N_t<200), the NSGA-II term usually dominates. Importantly, all terms are polynomial in the number of objectives and decision variables.
>
> By contrast, the computational complexity for qEHVI is O(mN_t^3+mK), where O(mN_t^3) is for the GP training and O(mK) is for the box decomposition of the Pareto front. The number of hyperrectangles K is super-polynomial in m. This is precisely why qEHVI becomes intractable for m>4.
>
> **Questions:**
>
> **1. Single Thompson sample.** Please check our response to reviewer **u5uz** for detailed discussion.
>
> **2. Clipping in Eq. 10.**
>
> We interpret this concern as indicating that clipping anchors the constraint to an already-evaluated point, potentially causing over-exploration of a known (limited) region. Our clipping rule is designed as a numerical stabilizer: it is triggered precisely when the sampled target exceeds all current observations on that objective — meaning the clipped threshold is set to the current maximum observed value. The resulting feasible region is highly uncertain; except for the boundary-defining observed point, there are typically no evaluated solutions in that region. Far from over-exploration, cEI is supposed to explore this highly uncertain region naturally.
>
> In practice, clipping arises primarily in early iterations when the posterior is uncertain and the sampled front overshoots the currently attainable region. As the number of observations grows and the posterior concentrates, the sampled front increasingly aligns with the true front and clipping is triggered less frequently.
>
> We conducted an ablation study comparing STAGE-BO with and without clipping: https://anonymous.4open.science/r/ICML-1EEC/Ablation_clip.md . Results show that on most benchmarks the two variants perform comparably, confirming that clipping acts primarily as a stabilizer. On a subset of benchmarks, clipping leads to measurable improvements, suggesting that the larger feasible region induced by clipping benefits optimization.
>
> **3. Scalability.** Please check our response above regarding runtime complexity.

---

> > ### Author Rebuttal · Reviewer_XQoK · 2026-04-03
> >
> > I have no further major concerns. However, I am still somewhat uncertain whether the significance of this work is sufficient for presentation at this conference. I will use the discussion phase to further assess its significance and then consider whether to raise my score.

---

> > > ### Author Response · Authors · 2026-04-06
> > >
> > > Thank you for the update - we are glad that the earlier concerns have been resolved. We would like to briefly restate our contributions: a highly performant, flexible, and efficient method for finding Pareto fronts with superior coverage by treating the multi-objective optimization as a problem of setting the constraints for single-objective optimization.
> > >
> > > Specifically, STAGE-BO introduces an efficient optimization principle for MOBO: instead of optimizing HV directly, it explicitly identifies under-covered regions of the Pareto front and converts MOO into a sequence of adaptive inequality-constrained single-objective subproblems to fill in these gaps. It removes the need for HV computation and also avoids the implicit bias of HV-based methods towards knee regions.
> > >
> > > Moreover, STAGE-BO is highly flexible. It extends naturally, without changing its core mechanism,  to unconstrained, constrained, and preference-aware MOBO. The central framework — identify a target gap, induce an adaptive epsilon-constraint problem around that target, and solve the resulting constrained BO subproblem — is also not tied to specific algorithmic choices such as Thompson sampling, NSGA-II, or cEI, making the contribution broader than a single solver combination (please check our response to Reviewer **J1pk** for detailed discussion). Our extensive experiments across extensive benchmark problems — spanning six synthetic and eight real-world tasks— demonstrate that STAGE-BO is highly competitive overall, with especially strong Pareto-front coverage (IGD) and competitive or strong HV performance.
> > >
> > > As we discussed in the Introduction and Related Work sections, recent diversity-guided MOBO methods [1–4] reflect growing interest in recovering not only high-quality Pareto solutions, but also well-covered tradeoff sets. We believe STAGE-BO makes a principled contribution to this direction: rather than treating diversity as an auxiliary goal alongside HV, it makes geometric gap-filling the main acquisition principle. In this way, it achieves strong performance in terms of both IGD and HV, while avoiding the need for HV computation.
> > >
> > > We hope this clarifies why we view the contribution as significant, and we thank the reviewer again for the thoughtful discussion.
> > >
> > >
> > > [1] Konakovic Lukovic, M., et al. (2020). Diversity-guided multi-objective Bayesian optimization with batch evaluations. NeurIPS.
> > >
> > > [2] Ahmadianshalchi, A., et al. (2024). Pareto front-diverse batch multi-objective Bayesian optimization. AAAI.
> > >
> > > [3] Renganathan, S. A., et al. (2025). qpots: Efficient batch multiobjective Bayesian optimization via pareto optimal thompson sampling. AISTATS.
> > >
> > > [4] Ngo, L., et al. (2025). MOBO-OSD: Batch Multi-Objective Bayesian Optimization via Orthogonal Search Directions. NeurIPS.

---

### Official Review · Reviewer_u5uz · 2026-03-10

**Soundness:** 3
**Presentation:** 3
**Significance:** 3
**Originality:** 3
**Overall Recommendation:** 5
**Confidence:** 4

**Summary:**

The paper proposes a new method for multi-objective Bayesian optimization that converts Pareto optimization into a sequence of constrained optimization problems. The core idea is to identify the largest gap in the Pareto front, and then construct a set of constraints that will fill that gap.The method applies NSGA-II to a posterior sample of the function (via spectral approximation) to compute a posterior sample of the Pareto front. It then solves a maximin problem to find the places in each sampled front that is furthest from the current data, and lastly constructs a constrained optimization problem that targets specifically that segment of the Pareto front. That problem is solved with regular constrained BO (constrained EI). The method performs well on real problems, and does an especially good job of improving coverage of the Pareto front over baselines.

**Compliance With Llm Reviewing Policy:**

Affirmed.

**Final Justification:**

The rebuttal addressed my main concern and, for the reasons outlined in the original review, I find the paper to be interesting, original, and useful and favor acceptance.

**Key Questions For Authors:**

1. How was qEHVI adapted to the preference-aware setting (Section 5.3)?

**Limitations:**

yes

**Strengths And Weaknesses:**

Strengths:
1. Overall, the paper solves an important problem in a novel, interesting, and well-motivated way.
2. The illustration in Fig. 1 provides a compelling motivation for a method that seeks to maximize coverage rather than hypervolume alone.
3. The paper is well written, and I was able to follow the development of the method without any issue.
4. All aspects of the method are well-motivated, and the method is free of hyperparameters that require tuning.
5. The empirical evaluation is thorough, with an appropriate set of benchmarks and problems.
6. The method does perform as expected given its motivation.
7. The set of ablations in the appendix are comprehensive and informative.

Weaknesses:
1. No description is given on how qEHVI was adapted to the preference-aware setting (Section 5.3). The current thing to do is to set the reference point to the corner of the ROI hypercube, so that positive hypervolume is only accumulated within the ROI. Was this done? If not, it needs to be done.
2. The construction of the optimization problem is done using a single sample from the Pareto front. It seems like some additional robustness and performance could be gained by using multiple samples and more fully approximating the posterior of the Pareto front.

Minor:
1. Missing period to end sentence in L066 left column.
2. L159 right column, missing S in "tandard".
3. L322 right column, typo "independence" --> "independent"

---

> ### Author Rebuttal · Authors · 2026-03-31
>
> Thank you for the positive and thoughtful assessment. We answer your concerns below.
>
> **Weaknesses and questions**
>
> **How qEHVI was adapted to the preference-aware setting** Thanks for raising this important point. In our paper, we adaptively set the reference point for qEHVI, because simply setting the reference point at the corner of the ROI hypercube is not always appropriate.  As illustrated in Fig 3 of our paper on page 6, the user-defined ROI may be overly ambitious and lie entirely beyond the attainable front. In this case, setting the reference point at the corner of the ROI hypercube means no observed solution dominates the ROI corner, and qEHVI accumulates zero hypervolume improvement throughout the entire optimization and becomes ineffective.
>
> To avoid this, our implementation shown in the paper adapted the qEHVI reference point using the observed Pareto front: we identified the observed Pareto front points lying within the feasible preference region (following the boundary-anchor logic defined in Fig 3) and then set the reference point to the coordinate-wise lower bounds of those points. If no observed point fell within this region, we used the coordinate-wise lower bounds of the entire observed Pareto front instead. However, this can still be conservative, especially in early iterations when observations are sparse and may not yet cover the preferred region.
>
> Given that qEHVI is sensitive to reference point, we further improved this adaptation and reran qEHVI in a way more consistent with our framework. We construct the qEHVI reference point from a Thompson-sampled path rather than from the observed front alone: we sample a posterior path over the design space, identify sampled points that fall inside the feasible preference region, and set the reference point to their coordinate-wise lower bounds. If no sampled point falls inside the preferred region, we fall back to the coordinate-wise lower bounds of the full sampled path.
>
> The updated results can be found here: https://anonymous.4open.science/r/ICML-1EEC/Update_roi_results.md With a better design reference point, the performance of qEHVI has been improved greatly but our method still achieves better HV, IGD, IGD+ and FD. More broadly, these results also highlight that qEHVI is quite sensitive to reference-point design in preference-aware settings, whereas STAGE-BO does not require choosing a reference point.
>
> We will clarify these in revision and publish our code.
>
> **Multiple instead of a single Thompson sample** We appreciate this suggestion. Since our predicted Pareto front is obtained by combining a Thompson sample with NSGA-II, we see two natural multi-sample variants.
>
> The first one is to draw multiple Thompson samples, average them, and run NSGA-II on the resulting mean curve. However, averaging multiple samples approximates the posterior mean. This interpretation would have the negative consequence that it would reduce the exploration behaviour, which is an essential element of Thompson sampling. Indeed, in an ablation study of replacing the Thompson sample with posterior mean:  https://anonymous.4open.science/r/ICML-1EEC/Ablation_posterior_mean.md,  the results are worse than with the single-sample approach. This is expected: the posterior mean is overly greedy, suppressing the uncertainty-driven variability that makes Thompson sampling effective for exploration.
>
> The second one is to run NSGA-II independently on each of multiple Thompson samples, generating multiple predicted Pareto fronts and yielding multiple candidate target points, and then aggregate them. While conceptually appealing, this raises two practical concerns. First, while we are more efficient than HV-based methods, NSGA-II is usually the most computationally expensive component in the early stage of STAGE-BO (the detailed runtime analysis can be found in the response to reviewer XQoK); running it multiple times per iteration would multiply this cost, partially undermining our computational efficiency advantage over HV-based methods. Second, it is unclear how to aggregate multiple target points into a single epsilon-constraint: taking their mean does not generally correspond to a point on any sampled Pareto front, and may produce constraints that define an empty feasible region.
>
> We believe the single-sample design strikes a good balance between exploration, computational efficiency, and constraint validity. One Thompson sample is also commonly adopted in MOO literature [1-2].
>
> **Minor issues.** Thank you for catching typos; we will correct these in the revision.
>
>
> [1] Bradford, E., Schweidtmann, A. M., & Lapkin, A. (2018). Efficient multiobjective optimization employing Gaussian processes, spectral sampling and a genetic algorithm.
>
> [2] Renganathan, S. A., & Carlson, K. E. (2025). qpots: Efficient batch multiobjective bayesian optimization via pareto optimal thompson sampling.

---

> > ### Author Rebuttal · Reviewer_u5uz · 2026-04-03
> >
> > I thank the authors for their response and for the additional results they provided.
> >
> > Preference-aware qEHVI: Yes my experience is also that qEHVI is quite sensitive to the selection of the reference point, hence the importance of doing that in the best possible way for fair comparison, which the new results clearly do, so thank you for those.
> >
> > I think the paper introduces a useful and novel approach for a real and important problem, and so continue to favor its acceptance.

---

> > > ### Author Response · Authors · 2026-04-06
> > >
> > > Thank you very much for your thoughtful and supportive comments. We are glad that the additional results addressed the concerns. We sincerely appreciate your positive assessment of the novelty and usefulness of our approach.

---

### Official Review · Reviewer_yvUL · 2026-03-12

**Soundness:** 4
**Presentation:** 4
**Significance:** 4
**Originality:** 3
**Overall Recommendation:** 4
**Confidence:** 3

**Summary:**

This paper proposes a new MOBO method, particularly those based on HV maximization called STAGE-BO. This method moves away from HV calculations and instead utilizes the $\epsilon$-constraint principle, which converts a multi-objective problem into a series of single-objective sub-problems. This paper uses Thompson Sampling and evolutionary algorithms (like NSGA-II) to estimate the current approximate Pareto front from the posterior distribution.

The algorithm identifies the largest "geometric gaps" (under-explored regions) in the current Pareto front approximation. These gaps are used to set dynamic $\epsilon$-constraint thresholds. The problem is then solved using the Constrained Expected Improvement (CEI) acquisition function to specifically target and fill those gaps.

The method was tested on synthetic benchmarks (ZDT, DTLZ) and real-world engineering problems (Rocket Injector design, Water Planning). STAGE-BO significantly outperforms baseline methods (like qEHVI) in terms of Inverted Generational Distance (IGD), often by an order of magnitude, indicating a much more uniform distribution of solutions.Efficiency: By avoiding expensive HV computations, it scales efficiently to problems with a high number of objectives (e.g., 5 or 6 objectives). It achieves state-of-the-art results in both diversity of solutions and hypervolume convergence.

STAGE-BO provides a scalable, easy-to-implement, and mathematically grounded framework for MOO that ensures a diverse and well-distributed set of Pareto-optimal solutions while remaining computationally efficient for many-objective problems.

**Compliance With Llm Reviewing Policy:**

Affirmed.

**Ethical Review Concerns:**

NA.

**Ethics Expertise Needed:**

["Other Expertise"]

**Final Justification:**

keep my original score.

**Key Questions For Authors:**

1. Line 212, right. NSGA2 is developed in around 2002, not 2012.

2. Too many bold fonts. Descrease some of them will make the paper more professional.

3. How does the "Target Identification" step work? How do Thompson Sampling and the internal evolutionary algorithm (like NSGA-II) collaborate to estimate the unseen parts of the Pareto front?

4. What is the specific logic behind "Gap-Filling"? How does STAGE-BO mathematically define a "geometric gap"? Specifically, how does it use the concept of Fill Distance to quantify the space between current observations and the predicted Pareto front?

5. How is "Preference-Awareness" implemented? If a decision-maker is only interested in a specific Region of Interest (ROI) on the Pareto front, what parameters or boundaries need to be adjusted in the STAGE-BO framework?

**Limitations:**

1. Not enough baselines. consider to cite Hypervolume-Guided Decomposition for Parallel Expensive Multiobjective Optimization / MOEAD-expensive or some thing related.

**Strengths And Weaknesses:**

1. This work use FD distance, which is developed in Zhang 2024, which very natural at least to me.

2. FD distance is a natural extension for MOBO problem which aim for uniform designs.

3. The testing problems are still too easy. RE problems have been developed for many years, consdier use some new tasks like real world probelms.

---

> ### Author Rebuttal · Authors · 2026-03-31
>
> Thank you for the positive assessment and for recognizing the core contribution. We answer your concerns below:
>
> **Weaknesses:**
>
> **Real-world tasks.** We further expanded our empirical evaluation with an additional real-world application: hyperparameter optimization for privacy-preserving machine learning. This complements our existing study on six synthetic benchmarks and eight real-world problems commonly used in the MOO literature. In this new task, we optimize five hyperparameters—batch size, learning rate, noise level, clipping norm and epoch—under two objectives: model privacy and accuracy. Results are available here: https://anonymous.4open.science/r/ICML-1EEC/Real_world_application.md   Since the true Pareto front is unknown in this setting, IGD cannot be computed. Our method consistently achieves the best HV performance.
>
>
> **Questions:**
>
> **1-2.** Thank you for catching the NSGA-II date typo, we will correct it, and reduce use of boldface.
>
> **3 How target identification works.** In each BO iteration, first we draw one Thompson sample from the GP posterior for each objective. This yields a sampled multi-objective function, which represents one plausible realization of the unknown true objectives under the current posterior belief. Crucially, this sampled function is defined over the entire input domain, not just at observed points — it therefore extrapolates the objective landscape into regions that have not yet been evaluated.
>
> Second, we run NSGA-II on this sampled function to generate its Pareto front. Since the sampled function is a cheap-to-evaluate surrogate, NSGA-II can explore the full input space efficiently. The resulting sampled Pareto front thus covers both observed regions and as-yet-unevaluated regions of the objective space that are plausible under the current posterior.
>
> Target identification is then performed on this sampled front: we select the point​ on the sampled front that maximizes the minimum distance to current observations (the maxmin fill-distance criterion, Eq. 8). This identifies the largest geometric gap in the current Pareto front approximation.
>
> Intuitively, Thompson sampling extrapolates plausible unseen front geometry, while NSGA-II serves as the inner optimizer used to expose that geometry. Thus, Thompson sampling provides uncertainty-aware front prediction, and NSGA-II is the numerical procedure used to extract an approximate Pareto front from that prediction. We visually show this process here https://anonymous.4open.science/r/ICML-1EEC/Illustration.md  .
>
>
> **4 Gap-filling logic.** STAGE-BO defines a geometric gap in the objective space, grounded in the Fill Distance criterion (Eq. 8). As discussed above, Thompson sample and NSGA-II together generate a discrete sampled Pareto front, on which we identify the target point that maximizes the minimum distance to the current observations. This represents the largest uncovered region on the predicted front. By repeatedly selecting the farthest uncovered Pareto point as the target, the algorithm explicitly reduces the largest coverage gap.
>
> Once the target is selected, we convert it into an adaptive epsilon-constraint subproblem: one objective is optimized directly, while the remaining objectives are required to satisfy threshold values derived from Eq. 10. cEI is then used to search for a new design likely to improve the chosen objective while satisfying the thresholds, thereby filling that gap.
>
> In short: Thompson sampling + NSGA-II predict where the front likely is, FD identifies where the largest gap is, the epsilon-constraint encodes what filling that gap means, and cEI finds which input is most likely to achieve it.
>
> **5 Preference-awareness.** Preference-awareness is implemented by restricting target selection to a user-specified region of interest (ROI) in the objective space. The only modification to the standard pipeline is in the cheap inner MOO problem (Eq. 19): when using Thompson sampling + NSGA-II to predict the sampled Pareto front, the user's preference bounds are added as constraints, so that only front regions satisfying the ROI are considered. The maxmin gap search then operates on this ROI-restricted front, and the resulting epsilon-constraints are derived from that localized target point. The acquisition function (cEI, Eq. 11) remains completely unchanged.
>
> In other words, the max-min gap search is localized to the preferred region, and the resulting epsilon-constraints are generated from that ROI-restricted target point. This allows the method to focus evaluations on decision-relevant parts of the front.
>
> **Limitations:**
>
> **Baselines.** We appreciate this suggestion. We ran two decomposition-based baselines mentioned by reviewers: MOEAD-EGO and DirHV-EGO. The results can be found here: https://anonymous.4open.science/r/ICML-1EEC/New_baselines.md . STAGE-BO is consistently better than the two baselines.

---

> > ### Author Rebuttal · Reviewer_yvUL · 2026-04-01
> >
> > I thank the authors for their rebuttal. Some of my concerns are addressed, and I am keeping my score unchanged.

---

> > > ### Author Response · Authors · 2026-04-06
> > >
> > > Thank you for your update. We are glad that our rebuttal addressed some of your concerns. We would be happy to continue the discussion and do our best to further address the remaining issues.

---

### Decision · Program_Chairs · 2026-04-30

**Decision:**

Accept (regular)

**Comment:**

This is a borderline paper with divergent scores ranging from weak reject to accept. The submission proposes a technically sound method for multi-objective Bayesian optimization that efficiently targets under-explored regions of the Pareto front without expensive hypervolume computations.

During the rebuttal, the authors significantly strengthened the paper by providing additional experiments, including a new real-world privacy-preserving machine learning task and comparisons against requested decomposition-based baselines.

While two reviewers maintained their weak reject scores, primarily questioning the practical significance, the proposed framework is mathematically grounded, well-evaluated, and provides a highly scalable alternative to existing hypervolume-based methods. Because the work is technically solid, non-redundant, and useful to the conference community, I recommend weak acceptance.